# AgentWebBench: Benchmarking Multi-Agent Coordination in Agentic Web

**Shanshan Zhong** [1]   **Kate Shen** [2]   **Chenyan Xiong** [1]

## Abstract

Agentic Web is an emerging paradigm where autonomous agents help users use online information. As the paradigm develops, content providers are also deploying agents to manage their data and serve it through controlled interfaces. This shift moves information access from centralized retrieval to decentralized coordination. To study this setting, we introduce AgentWebBench, a benchmark that evaluates how well a user agent synthesizes answers by interacting with website-specific content agents. We evaluate four tasks that cover common web information needs, spanning ranked retrieval (web search, web recommendation) and open-ended synthesis (question answering, deep research). Across seven advanced LLMs and three coordination strategies, multi-agent coordination generally lags behind centralized retrieval as expected, because user agent cannot directly access the corpus, but the gap shrinks with model scale and can even outperform centralized retrieval on question answering. This benchmark also enables us to study properties of the emerging paradigm of the digital world. We find that decentralized access concentrates traffic toward a small set of websites, test time scaling improves both interaction reliability and task performance, and strong results require sufficient interactions guided by careful planning. Finally, our failure analysis suggests that user agents need better planning and answer synthesis, while content agents need more reliable retrieval and evidence quality. Code, data, and APIs are released on https://github.com/cxcscmu/AgentWebBench.

## 1. Introduction

An increasing number of users are relying on large language model (LLM)-driven intelligent agents for accessing infor-

[1]Language Technologies Institute, School of Computer Science, Carnegie Mellon University [2]Anaxi Labs. Correspondence to: Shanshan Zhong <szhong2@cs.cmu.edu>.

*Proceedings of the $43^{rd}$ International Conference on Machine Learning*, Seoul, South Korea. PMLR 306, 2026. Copyright 2026 by the author(s).

mation, such as those behind ChatGPT (OpenAI, 2025a) and Google AI (Google, 2024). This reflects an emerging trend toward Agentic Web (Yang et al., 2025d), a transformative paradigm where autonomous agents act on behalf of users to accomplish complex tasks across the internet. In this setting, LLM-based agents plan, navigate, and synthesize information from diverse web sources. They reduce the effort of information access and have been discussed as a common way users will interact with the web (Kanoulas et al., 2025; Wang et al., 2024).

However, agentic access is also reshaping the web toward more compartmentalized access, and its impact on users and content providers remains unclear. As agentic access grows, content providers have less incentive to expose raw data and increasingly restrict direct access. They expose information through APIs or agent-facing interfaces, often mediated by standardized protocols such as Model Context Protocol (MCP) (Anthropic, 2024), its web-facing extension WebMCP (Web Machine Learning Community Group, 2026), and related frameworks (Google Developers Blog, 2025; Lù et al., 2025; Tarbert, 2025; Kong et al., 2025). At the same time, high-profile legal disputes against AI companies over the use of copyrighted data for model training (Bondari, 2025; Bannigan et al., 2025) further incentivize providers to retain tighter control over their content. Together, these developments suggest a broad shift in how the web is accessed, toward provider-controlled interfaces and multi-agent interactions. In some regions, users already access much online content through closed platform ecosystems (e.g., mini-apps, TikTok) rather than open webpages (Schreieck et al., 2023).

Understanding how this shift reshapes the web as a multi-agent ecosystem is therefore critical and timely. To benchmark and understand this emerging shift in web access, and to support future Agentic Web development, we introduce AgentWebBench as shown in Fig. 1. AgentWebBench formalizes the Agentic Web as a decentralized information ecosystem. Given a user intent, a user agent selects relevant websites, queries their content agents through agent-facing interfaces, and synthesizes the returned evidence into final answers. Selected content agents retrieve information within their own domains and respond with evidence to the user agent. We evaluate four information-seeking tasks that cover common web needs, including ranked retrieval (web

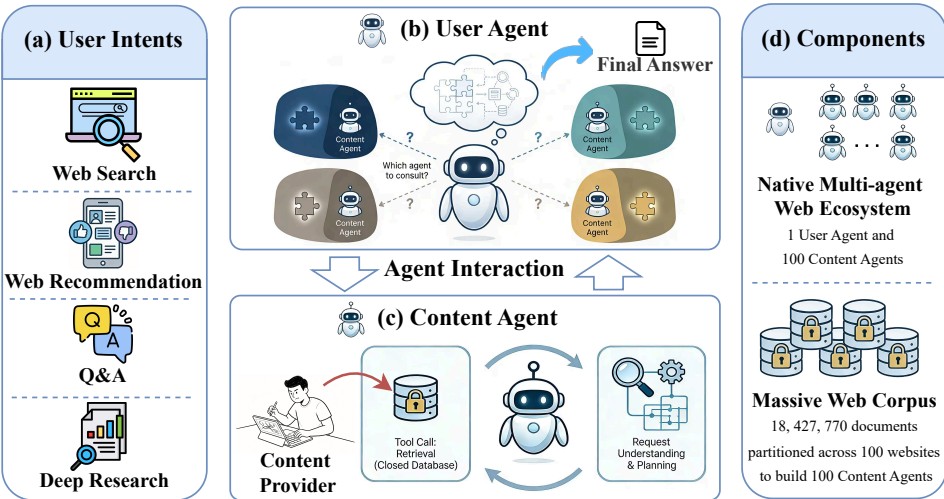

*Figure 1.* Overview of AgentWebBench. The benchmark models the Agentic Web paradigm where user agents coordinate with multiple autonomous content agents across information-seeking tasks. (a) User intents spanning four representative tasks: web search, web recommendation, question answering, and deep research. (b) User agent that analyzes intents, selects and queries content agents, and synthesizes responses. (c) Content agents that process requests and retrieve information from their proprietary domains. (d) Environment Components comprising 100 websites with 18.4M documents, each managed by an independent content agent.

search, web recommendation) and open-ended synthesis (question answering, deep research).

The benchmark includes 100 websites with 18.4 million documents across diverse domains, reflecting complex real-world web information sources. Each website is served by a content agent that accesses its own document corpus. Furthermore, to understand what drives performance in this Agentic Web setting, we compare three coordination strategies. These strategies progressively introduce model reasoning for website selection and agent-to-agent communication: (a) $Tool_E$ selects websites via embedding similarity and retrieves documents through tools, without agent communication; (b) $Tool_P$ uses LLM reasoning for website selection but still relies on tool-based retrieval; and (c) Multi-Agent enables agent-to-agent communication where both user and content agents reason autonomously. Under each strategy, we evaluate agents powered by seven advanced LLMs. This provides a comprehensive baseline for decentralized information access in Agentic Web.

Our experiment results show a general performance gap between decentralized coordination and centralized web retrieval. This gap narrows substantially as model scale increases. For certain tasks such as question answering, multi-agent coordination can outperform the centralized baseline, since iterative evidence gathering fits the agentic setting. These findings show that Agentic Web still faces effectiveness challenges, but performance can improve as models advance. Beyond task effectiveness, AgentWebBench also allows us to analyze key properties of this emerging paradigm. We focus on three dimensions that shape real-world use:

systemic impact, efficiency and optimization, and failure modes. First, we observe traffic concentration. While Agentic Web can improve reliability through content-side filtering, it often steers users toward a small subset of websites. This can reduce the discoverability of content providers and may limit information diversity. Second, we identify practical paths to improve performance. Allowing agents to think before acting improves both interaction reliability and task performance, suggesting that test-time scaling can narrow the performance gap. Our efficiency analysis further shows that effective coordination needs enough interactions. Finally, our failure analysis suggests different bottlenecks across agents. User agents need better planning and answer synthesis, while content agents need reliable retrieval and higher-quality evidence verification. Together, these findings improve our understanding of this emerging paradigm and provide a platform to study it systematically, pointing to clear opportunities for future progress of Agentic Web. In summary, our contributions are threefold:

- We introduce AgentWebBench, the first comprehensive benchmark for evaluating agent performance in Agentic Web across four common web tasks, providing a foundation for research in this emerging paradigm.
- Building a decentralized architecture where a user agent coordinates with multiple content agents, we show that the decentralized nature of Agentic Web makes it generally trail the centralized baseline. However, the gap narrows with model scaling and can reverse on question answering.
- To understand this emerging paradigm beyond end-task accuracy, we provide insights that characterize ecosystem impact and identify improvement pathways. Specifically,

we study traffic concentration, test-time scaling, interaction efficiency, and failure analysis.

## 2. Related Work

**Tool-using and API-based LLMs.** LLMs are shifting from text generators to autonomous agents. This shift matters for information access, where retrieval and synthesis of external knowledge are essential. Early frameworks such as ToolFormer (Schick et al., 2023) and ToolLLM (Qin et al., 2024) showed that models can invoke external tools, such as search engines and calculators, to extend their capabilities, and recent work further analyzes data-driven skill extensions of LLMs (Ling et al., 2026; Zhong et al., 2026). Retrieval-Augmented Generation (RAG) further grounds model outputs in real-time documents (Gao et al., 2023b; Salemi & Zamani, 2024). Advanced variants like FLARE (Jiang et al., 2023) and Self-RAG (Asai et al., 2024) enable iterative and self-reflective retrieval. These advances underpin agentic search and deep research systems (Li et al., 2025; Zheng et al., 2025). Such systems decompose queries, plan multi-source retrieval, and synthesize reports across scientific, code, and multimodal domains (Huang et al., 2025b; Ren et al., 2025; Masterman et al., 2024; Koh et al., 2024). Growing user adoption of agents for complex tasks reflects their effectiveness (Guo et al., 2024; Hong et al., 2025) and provides foundation for Agentic Web (Yang et al., 2025d).

**Agentic Web.** The capabilities above are converging to reshape internet architecture toward an Agentic Web (Yang et al., 2025d), where autonomous agents act on behalf of users to accomplish tasks. Several trends support this shift. First, users increasingly delegate complex tasks, such as travel planning (Chen et al., 2024a) and deep research (Huang et al., 2025b; Xi et al., 2025), to AI assistants. This reflects a preference for goal-oriented interaction over manual navigation (Guo et al., 2024; Hong et al., 2025; Genspark, 2025). Second, the web is becoming more compartmentalized. Content is often behind APIs and login walls, which requires agent-mediated access through structured interfaces (Yang et al., 2025c). Third, providers are designing for agent entry (Anthropic, 2024). Examples include initiatives like Natural Language Web (NLWeb) and the adoption of agent protocols such as MCP (Anthropic, 2024) and A2A (A2A Project, 2025). Conceptual work formalizes the Agentic Web as a distributed ecosystem of persistent, goal-directed agents that coordinate across web resources (Petrova et al., 2025; Lù et al., 2025; Chaffer, 2025). This work also frames a transition from linked documents to acting intelligences (Yang et al., 2023; Kapoor et al., 2024; Wang et al., 2023). As this paradigm evolves, content providers are increasingly deploying their own agents. This introduces coordination challenges for information access.

**Multi-Agent Systems and Benchmarks.** Multi-agent coordination has long been studied in AI. LLMs have renewed interest in this area through multi-agent frameworks like AutoGen (Wu et al., 2024), ChatDev (Qian et al., 2024), and AgentNet (Yang et al., 2025b). These systems use role specialization and agent communication to solve complex tasks. Besides, web benchmarks such as WebArena (Zhou et al., 2024), Mind2Web (Deng et al., 2023), and ST-WebAgentBench (Levy et al., 2026) evaluate autonomous task completion, safety, and trustworthiness, and complementary LLM benchmarks target capabilities such as routing across models (Huang et al., 2025c), creativity (Huang et al., 2025e), and benchmark compression (Huang et al., 2025d). However, they mainly assess single-agent performance and do not capture the coordination demands of emerging Agentic Web. In this setting, information access often requires coordinating with multiple autonomous agents due to access restrictions and role differentiation (Yang et al., 2025d). AgentWebBench addresses this gap by evaluating multi-agent coordination in a concrete architecture across common web tasks, including web search, web recommendation, question answering, and deep research (Wu et al., 2026; He et al., 2025; Zheng et al., 2025).

## 3. AgentWebBench

In this section, we introduce AgentWebBench (Fig. 1), a benchmark for evaluating multi-agent coordination of the emerging Agentic Web. We describe the agent types, coordination strategies, tasks, and evaluation environment.

### 3.1. Agent Types

AgentWebBench models information access through a decentralized architecture. Information is organized by website domains, and each website maintains its own retrieval infrastructure. AgentWebBench includes two agent types:

**User agent.** Acts on behalf of users, processes queries, and coordinates information gathering across multiple websites. It has no direct corpus access and must interact through website-specific interfaces, reflecting realistic access constraints in the Agentic Web setting.

**Content agents.** Each operates autonomously for a specific website domain. It performs retrieval within its domain and returns both natural language summaries and document contents. Summaries support high-level reasoning, while full contents enable downstream processing such as ranking, evidence aggregation, and answer synthesis.

### 3.2. Coordination Strategies

We design three coordination strategies that progressively introduce key capabilities of the Agentic Web paradigm. All strategies share the same decentralized architecture but differ in: (a) website selection: how relevant websites are

identified, and (b) document retrieval: whether documents are obtained through automated tools or agent interactions.

### 3.2.1. TOOL-BASED EMBEDDING SELECTION (TOOL$_E$)

This strategy is a lightweight baseline that relies on automated tools and does not use agent communication.

**Website selection.** Performed automatically via embedding similarity. The query embedding is compared with precomputed website embeddings to select the top $k$ websites.

**Document retrieval.** For each selected website, a dense retrieval tool is called to obtain candidate documents. No content agents are involved.

### 3.2.2. TOOL-BASED PROMPTED SELECTION (TOOL$_P$)

This strategy uses LLM reasoning for website selection while keeping tool-based document retrieval, as in Tool$_E$.

**Website selection.** The user agent receives descriptions of all websites in its prompt and uses LLM reasoning to select relevant ones for the query.

**Document retrieval.** Given the query and selected websites, the user agent issues a website-document retrieval tool call using the same dense retrieval tool as Tool$_E$.

### 3.2.3. MULTI-AGENT COORDINATION

This strategy enables agent-to-agent communication. Both user and content agents act autonomously, with independent reasoning and decision-making.

**Website selection.** The user agent plans multi-turn interactions and dynamically selects content agents based on evolving information needs. It uses conversational feedback as additional signals beyond initial query. Website descriptions are included in agent's prompt to guide selection.

**Document retrieval.** Selected content agents perform retrieval within their own domains using their own prompts, generate natural language summaries, and return structured responses backed by in-domain documents. This peer-to-peer coordination enables iterative refinement and adaptive information gathering beyond purely tool-based approaches.

### 3.3. Tasks and Evaluation

AgentWebBench supports four tasks that represent common web information needs, spanning ranked retrieval and open-ended synthesis. Web search and web recommendation cover two canonical retrieval settings, ad-hoc query retrieval and history-conditioned retrieval. Question answering and deep research require multi-step evidence gathering and cross-source synthesis. Together, these tasks let us evaluate coordination behaviors for both retrieval and synthesis in the Agentic Web. Table 1 summarizes each task, dataset

*Table 1.* Summary of tasks, datasets, and evaluation metrics.

| Task | #Sample | Output | Metrics |
|---|---|---|---|
| Web Search | 354 | Ranked documents | NDCG@$k$, Recall@$k$ |
| Web Recommendation | 281 | Ranked documents | NDCG@$k$, Recall@$k$ |
| Question Answering | 53 | Short answer | Accuracy, F1 |
| Deep Research | 331 | Long report | KPR, KPC, Clarity, Insight |

size, and metrics. See Appendix J for more details.

### 3.3.1. WEB SEARCH

Given a query $q$, the task requires returning a ranked list of document identifiers $\mathcal{D} = \{d_1, d_2, \ldots, d_k\}$ through coordination with content agents, each providing access to a single website's documents. This task isolates coordination for source selection and high-recall evidence gathering under strict website boundaries.

**Dataset.** We start from the MS MARCO Web Search test set (Chen et al., 2024b) and keep 354 queries whose ground-truth documents are covered by the AgentWebBench corpus.

**Evaluation.** Standard information retrieval (IR) metrics (Dai et al., 2023): NDCG@$k$ (N@$k$) and Recall@$k$ (R@$k$).

### 3.3.2. WEB RECOMMENDATION

Given a user browsing history $\mathcal{H} = \{d_1, d_2, \ldots, d_n\}$, the task requires inferring the user's next intent and recommending relevant documents through coordination between the user agent and content agents. This models personalized information access in real-world systems such as browser suggestions and online shopping recommendations (He et al., 2025; Huang et al., 2025a). This task stresses intent inference and proactive source selection.

**Dataset.** We adapt 281 user histories from ORBIT (He et al., 2025), retaining only instances whose target documents appear in the AgentWebBench corpus.
**Evaluation.** Ranking metrics: N@$k$ and R@$k$.

### 3.3.3. QUESTION ANSWERING

Given a query, the task requires producing a concise answer through multi-step retrieval and reasoning. Many queries require aggregating evidence from multiple closed-domain sources when no single content agent contains a complete answer. This requires strategic content agent selection and progressive answer synthesis through iterative interactions. This task emphasizes iterative evidence integration and faithfulness under decentralized access.

**Dataset.** We start from the short-answer subset of Deep-ResearchGym (Coelho et al., 2025) and retain 53 queries whose evidence is covered by the AgentWebBench corpus.

**Evaluation.** Correctness is measured by LLM-as-Judge accuracy and token-level F1.

### 3.3.4. DEEP RESEARCH

Given an open-ended query $q$, the user agent generates a comprehensive report $\mathcal{R}$ through iterative planning, multi-

step retrieval, and evidence synthesis across multiple sources. This task highlights long-horizon planning, information coverage, and structured report generation.

**Dataset.** We adapt 331 queries from DeepResearch-Gym (Coelho et al., 2025), retaining those whose supporting documents exist in the proposed AgentWebBench corpus.

**Evaluation.** We follow the metrics of DeepResearchGym. Report relevance is measured by Key Point Recall (KPR, coverage of salient points) and Key Point Contradiction (KPC, factual inconsistencies). Report quality is assessed via LLM-as-Judge on Clarity and Insight.

### 3.4. Web Corpus and Retrieval Infrastructure

To instantiate AgentWebBench in a reproducible setting, we construct an evaluation environment using a large-scale public web corpus with per-website retrieval infrastructure.

**Web Corpus.** AgentWebBench uses the English subset of ClueWeb22-B (Overwijk et al., 2022), a large-scale web corpus collected in 2022 that approximates the most frequently visited portion of the web. ClueWeb22-B contains approximately 200 million web pages, including around 87 million English pages. These pages are sampled by importance scoring to reflect real-world information needs, with low-quality and spam content filtered. We select the 100 websites with the largest document counts, yielding a decentralized set of sources that collectively contain 18,427,770 documents. Each website is assigned to one content agent.

**Content Agent Retrieval Infrastructure.** To enable efficient retrieval within each closed domain, we build an independent dense retrieval index for every website using the MiniCPM-Embedding-Light model (Coelho et al., 2025; MiniCPM Team et al., 2025), a strong dense retriever trained on large-scale query-document data. Each content agent maintains its own index independently and exposes retrieval capabilities through a tool call scoped to its website. This setup enforces realistic domain-specific access constraints, where content agents control their own information boundaries.

## 4. Experiment Methodologies

This section presents the experimental setup, including baselines, LLM configurations, and agent settings.

**Baselines.** We evaluate three coordination strategies (Tool$_E$, Tool$_P$, and Multi-Agent) and the following baselines:

- **Classical**: Direct retrieval from a centralized document index covering AgentWebBench corpora, representing centralized retrieval with universal access. This provides a reference for performance without access constraints.

- **Classic IR** (Gao et al., 2023a): Embedding-based dense retrieval for web search without any LLM agents.

**LLMs.** We evaluate agents powered by seven advanced LLMs (Yang et al., 2025a; DeepSeek-AI, 2025; OpenAI, 2025b; Google DeepMind, 2025), including Qwen3 series (4B, 14B, 30B-A3B, Next-80B-A3B (80B-A3B)), DeepSeek-V3.2, GPT-5-mini (GPT-5), and Gemini-3-Flash (Gemini-3). All LLMs adopt the thinking mode. More LLM implementation details are provided in Appendix D.

**Agent settings.** We set the maximum iterations to 15 turns per agent. Agent prompts and corpus information are provided in Appendix L and M, respectively.

## 5. Evaluation Results

In this section, we first report overall performance on AgentWebBench across LLMs and tasks. We then analyze multi-agent coordination from three angles: ecosystem impact, efficiency, and failure causes. Specifically, we ask three research questions:

- How does decentralized coordination change web traffic and source diversity? We analyze deep research because it involves diverse browsing and best reflects traffic shifts.
- How do test time scaling and interaction budget affect coordination performance? We report results across all tasks to obtain representative findings.
- Where do errors come from in the Agentic Web architecture, and why? We focus on tasks with clear ground truth (web search, web recommendation, and question answering) to enable consistent responsibility assignment.

### 5.1. Overall Performance

Table 2 summarizes five findings from our overall performance comparison on Agentic Web.

**(1) Overall performance is modest.** Results are modest across tasks and methods, which shows that AgentWebBench remains challenging under decentralized access. The main challenges are planning, selecting relevant content agents, and synthesizing evidence. On web search, the best LLM-based method (Gemini-3, Classical) only slightly improves over the Classic IR baseline. Web recommendation is harder, and most methods have near-zero recall (refer to Appendix I for more analysis of this task).

**(2) Website selection is a bottleneck.** We compare Tool$_E$ and Tool$_P$ to isolate website selection. They both remove content agents and only differ in how they choose websites. On web search, Tool$_P$ is often better than Tool$_E$, which suggests that LLM reasoning helps pick more relevant websites than embedding similarity. On deep research, Tool$_E$ stays competitive, which suggests retrieval can work well once the website set is roughly correct.

**(3) Content agents are less stable on retrieval-heavy tasks.**

*Table 2.* Performance comparison across four tasks, including web search, web recommendation, question answering (QA), and deep research. Metrics (%) are task-specific. Classic IR is a non-agent baseline; "-" indicates not applicable. **Bold** and underline indicate the best and second-best results, respectively.

| LLM | Method | Web Search | | | | Web Recommendation | | | | QA | | Deep Research | | | |
|---|---|---|---|---|---|---|---|---|---|---|---|---|---|---|---|
| | | N@3 | N@5 | R@3 | R@5 | N@3 | N@5 | R@3 | R@5 | Acc. | F1 | KPR | KPC↓ | Clarity | Insight |
| | Classic IR | 47.86 | 51.04 | 57.63 | 65.25 | - | - | - | - | - | - | - | - | - | - |
| Qwen3-4B | Classical | **47.65** | **47.65** | **54.80** | **54.80** | **0.80** | **0.80** | **1.07** | **1.07** | 18.87 | 21.43 | 52.74 | 1.48 | 71.45 | 63.84 |
| | Tool$_E$ | 30.67 | 31.03 | 36.44 | 37.29 | 0.00 | 0.00 | 0.00 | 0.00 | 13.21 | 15.21 | 53.17 | 2.19 | **77.13** | **74.35** |
| | Tool$_P$ | 32.62 | 33.21 | 35.31 | 36.72 | 0.58 | 0.58 | 0.71 | 0.71 | 13.21 | 16.13 | 47.83 | **1.37** | 67.79 | 61.99 |
| | Multi-Agent | 26.80 | 27.04 | 29.38 | 29.94 | 0.58 | 0.58 | 0.71 | 0.71 | 15.09 | 17.44 | 47.54 | 1.63 | 71.63 | 62.63 |
| Qwen3-14B | Classical | **47.59** | **47.59** | **55.37** | **55.37** | 0.22 | 0.22 | 0.36 | 0.36 | 16.98 | 16.73 | 54.40 | 1.29 | 71.57 | 59.12 |
| | Tool$_E$ | 25.82 | 27.94 | 30.23 | 35.31 | 0.18 | 0.18 | 0.36 | 0.36 | 20.75 | 17.90 | 54.62 | 0.68 | 75.83 | 65.92 |
| | Tool$_P$ | 30.06 | 31.12 | 33.90 | 36.44 | 0.00 | 0.14 | 0.00 | 0.36 | 20.75 | 13.18 | 49.12 | 1.13 | 63.11 | 52.81 |
| | Multi-Agent | 27.33 | 27.90 | 31.36 | 32.77 | **0.22** | **0.53** | 0.36 | **1.07** | **28.30** | **23.79** | 49.85 | 1.29 | 63.23 | 51.69 |
| Qwen3 30B-A3B | Classical | **49.90** | **49.90** | **57.34** | **57.34** | **0.36** | **0.36** | 0.36 | 0.36 | **28.30** | **26.77** | 54.99 | 2.59 | 76.74 | 69.88 |
| | Tool$_E$ | 30.24 | 30.97 | 35.03 | 36.72 | 0.18 | 0.18 | 0.36 | 0.36 | 22.64 | 25.71 | 57.95 | 2.94 | **81.54** | **79.73** |
| | Tool$_P$ | 32.24 | 33.29 | 36.44 | 38.98 | **0.36** | **0.36** | 0.36 | 0.36 | 24.53 | 24.92 | 54.82 | **2.14** | 73.05 | 68.37 |
| | Multi-Agent | 29.38 | 29.72 | 32.77 | 33.62 | 0.22 | 0.22 | 0.36 | 0.36 | 20.75 | 18.12 | 52.03 | 2.53 | 71.45 | 64.38 |
| Qwen3 80B-A3B | Classical | **48.65** | **48.78** | **54.80** | **55.08** | **0.18** | **0.18** | **0.36** | **0.36** | 33.96 | 33.29 | 59.11 | 2.16 | 77.40 | 68.70 |
| | Tool$_E$ | 27.32 | 27.92 | 29.94 | 31.36 | **0.18** | **0.18** | **0.36** | **0.36** | 35.85 | 35.56 | 62.90 | 2.20 | **85.62** | **82.69** |
| | Tool$_P$ | 33.73 | 34.30 | 37.85 | 39.27 | **0.18** | **0.18** | **0.36** | **0.36** | 32.08 | 31.63 | 57.85 | **1.92** | 74.05 | 65.86 |
| | Multi-Agent | 33.32 | 34.04 | 38.42 | 40.11 | 0.00 | 0.00 | 0.00 | 0.00 | **37.74** | **37.60** | 57.28 | 2.16 | 74.56 | 65.26 |
| DeepSeek V3.2 | Classical | **49.02** | **49.37** | **55.37** | **56.21** | 0.40 | 0.40 | 0.71 | 0.71 | 24.53 | 19.51 | 66.54 | 1.27 | 87.16 | 80.48 |
| | Tool$_E$ | 33.98 | 35.05 | 37.85 | 40.40 | 0.00 | 0.00 | 0.00 | 0.00 | 20.75 | 17.05 | 68.23 | **1.11** | **88.61** | **83.56** |
| | Tool$_P$ | 34.98 | 36.17 | 40.11 | 42.94 | **0.58** | **0.89** | 0.71 | **1.42** | 24.53 | 17.00 | 64.23 | 1.47 | 86.65 | 74.89 |
| | Multi-Agent | 33.42 | 34.23 | 39.27 | 41.24 | 0.18 | 0.18 | 0.36 | 0.36 | **30.19** | **29.55** | 64.95 | 1.59 | 87.07 | 75.77 |
| GPT-5 | Classical | **51.39** | **52.09** | **58.76** | **60.45** | **0.45** | **0.45** | **0.71** | **0.71** | 35.85 | 21.10 | 60.97 | **0.92** | 77.07 | 74.80 |
| | Tool$_E$ | 34.23 | 35.41 | 38.98 | 41.81 | 0.36 | 0.36 | 0.36 | 0.36 | **37.74** | 24.57 | 70.87 | 1.18 | 83.23 | 82.27 |
| | Tool$_P$ | 40.29 | 42.55 | 47.18 | 52.54 | 0.36 | 0.36 | 0.71 | 0.71 | 32.08 | **24.99** | 70.62 | 0.99 | 88.61 | 85.29 |
| | Multi-Agent | 40.14 | 42.30 | 46.61 | 51.69 | 0.00 | 0.14 | 0.00 | 0.36 | 33.96 | 22.78 | 69.39 | 1.28 | **89.15** | 84.44 |
| Gemini-3 | Classical | **52.21** | **53.77** | **60.17** | **63.84** | **0.58** | **0.73** | 0.71 | **1.07** | 60.38 | 56.15 | 63.58 | 1.67 | 83.56 | 76.86 |
| | Tool$_E$ | 39.56 | 40.49 | 44.63 | 46.89 | 0.36 | 0.36 | 0.36 | 0.36 | 56.60 | 51.92 | 65.17 | 1.44 | **89.37** | **86.16** |
| | Tool$_P$ | 47.52 | 49.07 | 56.50 | 60.17 | 0.45 | 0.59 | 0.71 | **1.07** | **67.92** | 60.26 | 62.38 | 1.61 | 80.60 | 71.90 |
| | Multi-Agent | 44.34 | 47.08 | 51.98 | 58.47 | 0.36 | 0.36 | 0.36 | 0.36 | **67.92** | **61.10** | 62.45 | **1.37** | 81.00 | 72.54 |

Tool$_P$ and Multi-Agent mainly differ in whether they use content agents. On web search, Tool$_P$ is usually stronger, which suggests the current content-agent retrieval strategy is not yet stable for retrieval-focused tasks. On deep research, the two methods are often comparable, so this issue is less visible in more generative settings.

**(4) Multi-agent coordination shows promising task-dependent performance.** We compare Multi-Agent with Classical to study the tradeoff of agent interactions under decentralized access. Multi-Agent is often lower on web search and deep research, but the gap is smaller for stronger models such as Gemini-3 and GPT-5. On question answering, Multi-Agent can be better than Classical, likely because iterative evidence gathering matches the task structure.

**(5) Model scale helps consistently.** Within the Qwen3 series, performance improves with model scale for both Multi-Agent and Classical. The trend is especially clear on coordination-intensive tasks such as question answering. These results suggest that stronger models can directly improve Agentic Web performance and further reduce the gap between decentralized and centralized access.

## 5.2. Concentration of Web Traffic

We study how the Agentic Web may reshape the web ecosystem by comparing traffic distributions. Fig. 2 shows a clear difference. In the Classical setting (yellow bars), citations are spread across many sources, ranging from academic repositories to community platforms. Under multi-agent coordination (blue bars), citations concentrate on a small set of domains (e.g., sciencedirect.com, wikipedia.org). This result reveals a key trade-off in coordination design. Decentralized access of Agentic Web acts as a strong filter. With better planning, the user agent may repeatedly select sources it deems most useful and ignore others. This can improve reliability, but it reduces the diversity of sources consulted. In contrast, the Classical centralized setting keeps more visibility for smaller and informal sources, such as educational forums or niche platforms. Overall, a shift toward an Agentic Web may further concentrate internet traffic and make it harder for most content providers to be discovered. More analysis can be found in Appendix A.

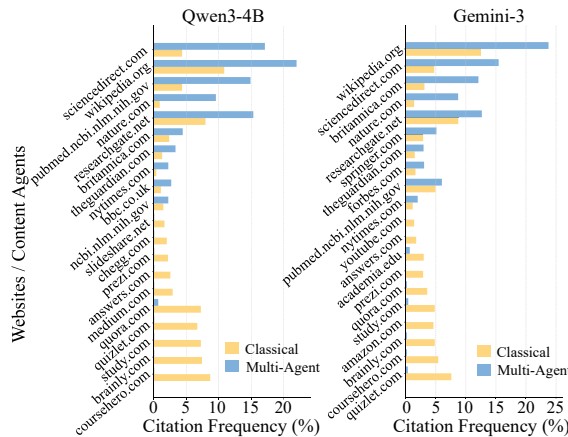

*Figure 2.* Impact of multi-agent coordination on web traffic. The charts show citation frequency for the 20 most cited sources (websites or content agents) for Qwen3-4B and Gemini-3 user agents on the deep research task. In each subplot, sources are ranked by the frequency difference (Multi-Agent minus Classical). The upper section highlights sources cited more often under multi-agent decentralized coordination, while the lower section shows sources cited more often in the Classical centralized setting.

### 5.3. Impact of Test Time Scaling

To understand how test time scaling affects coordination in the Agentic Web, we compare LLMs with and without thinking mode (standard instruction-tuned mode) on two tasks, web search and question answering, as shown in Fig. 3 (a,b). We analyze Qwen3-4B and Qwen3-30B-A3B because these two models have explicit thinking and nonthinking versions. Thinking mode consistently improves performance for both model scales, at the cost of a larger token budget. It lets agents simulate and verify action sequences before execution, leading to more deliberate planning. This matters in the Agentic Web, where user agents must choose which content agents to query, interpret responses, and decide when to stop searching. These decisions are hard to make with single-step inference. Thinking mode also improves interaction reliability by reducing malformed requests as shown in Fig. 3(c). This suggests that explicit reasoning acts as an internal check. Agents can verify that planned actions follow the protocol before producing outputs, which reduces hallucinations and protocol violations. Overall, test time scaling improves both performance and protocol adherence in the Agentic Web. Find more results in Appendix B.

### 5.4. Efficiency Analysis

We analyze interaction efficiency in AgentWebBench using four metrics as shown in Fig. 4: agent turns (number of responses), contacted content agents (#Agent, exploration breadth), retrieval requests (#Req., retrieval budget), and validity (fraction of successfully parsed requests or non-empty returns). Comparing the weakest and best models shows a clear pattern. Gemini-3 outperforms Qwen3-4B on most

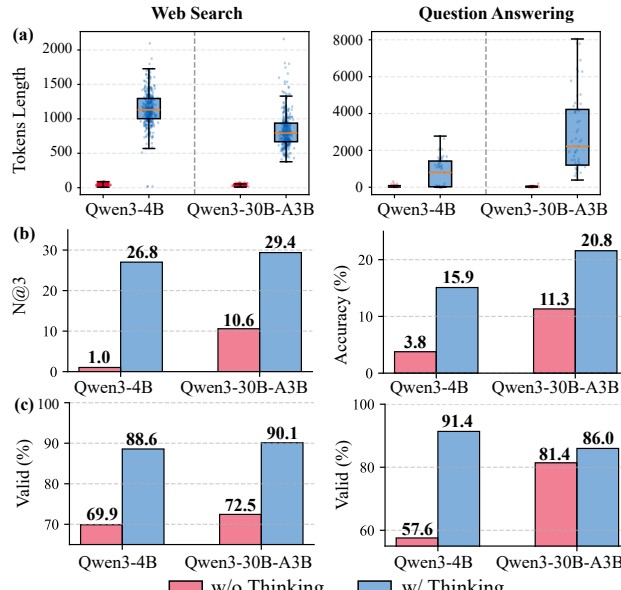

*Figure 3.* Effect of test time scaling on coordination performance. Valid (%) measures the fraction of successfully content-agent parsed requests.

*Table 3.* Failure mode (%) analysis of incorrect predictions. Each failure is attributed exclusively to either the user agent or the content agents based on task-specific rules. Percentages sum to 100% within each task.

| LLM | Web Search | | Web Rec. | | QA | |
|---|---|---|---|---|---|---|
| | User | Content | User | Content | User | Content |
| Qwen3-4B | 59.59 | 40.41 | 48.75 | 51.25 | 84.44 | 15.56 |
| Qwen3-14B | 64.83 | 35.17 | 62.59 | 37.41 | 84.21 | 15.79 |
| Qwen3-30B-A3B | 56.41 | 43.59 | 52.86 | 47.14 | 83.33 | 16.67 |
| Qwen3-80B-A3B | 54.90 | 45.10 | 60.85 | 39.15 | 93.94 | 6.06 |
| DeepSeek-V3.2 | 35.18 | 64.82 | 48.57 | 51.43 | 70.27 | 29.73 |
| GPT-5 | 42.21 | 57.79 | 60.22 | 39.78 | 68.57 | 31.43 |
| Gemini-3 | 30.66 | 69.34 | 56.52 | 43.48 | 94.12 | 5.88 |

tasks except web recommendation, and Gemini-3 agents always use more agent interactions and retrieval requests. In addition, Gemini-3 trades lower validity for broader coverage, while Qwen3-4B cannot turn higher validity into better results because it contacts fewer agents and issues fewer requests. These results suggest that efficiency in Agentic Web systems is not just about reducing interaction counts. Coordination strategies should decide when extra interactions are helpful and when they are redundant. This motivates adaptive interaction budgeting and planning-aware coordination. Find detailed results in Appendix C.

### 5.5. Failure Mode Analysis

We analyze where coordination fails in the Agentic Web with task-specific attribution on two discriminative tasks (web search and web recommendation) and one generative task (question answering). We exclude deep research because it has no exact ground-truth answers. For the two discriminative tasks, we assign responsibility deterministi-

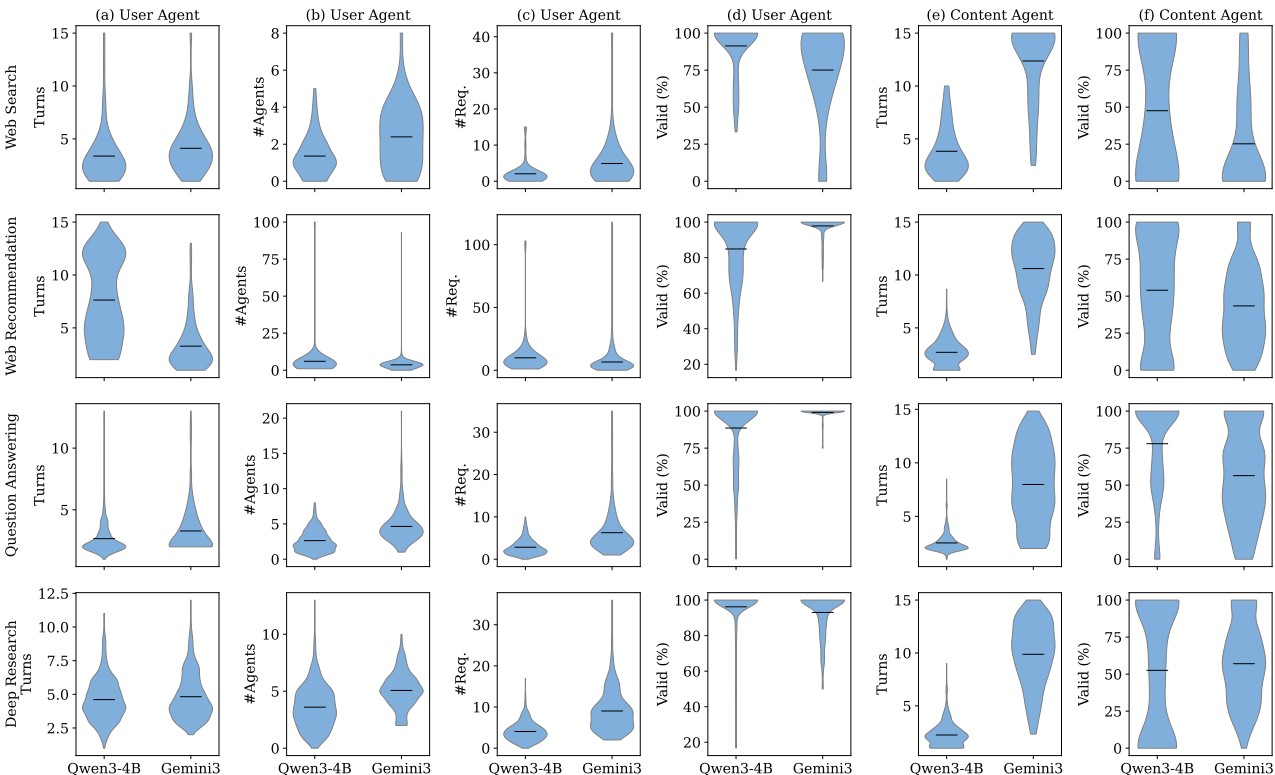

*Figure 4.* Interaction efficiency across tasks in AgentWebBench. Turn is the number of agent responses. #Agent and #Req. are the numbers of contacted content agents and retrieval requests. Valid (%) is the success rate of parsing requests or returning results.

---

**Failure on Question Answering (Gemini-3)**

**Query:** In Plato's analogy of the sun, what element does he compare to the Form of the Good in terms of its role in enabling knowledge?
**GT answer:** Light

**Search request:** search("Plato's analogy of the sun Form of the Good comparison", websites=[Wikipedia, Britannica])

**Content agent (key sentences):**
(1) **Wikipedia**: The Sun provides light, which allows the eye to see... Form of the Good provides truth and reality, allowing the soul to understand...

(2) **Britannica**: ...sun provides light that allows eyes to see... Form of the Good provides truth and reality, enabling the soul to know...

**User agent answer:** The Sun ×

---

cally. If the user agent never contacts a content agent that hosts any ground-truth document, we attribute the failure to the user agent. Otherwise, we attribute it to the content agents. For question answering, we measure evidence risk. An adjudication model (GPT-5-nano (OpenAI, 2025b)) judges whether the retrieved evidence from selected content agents could plausibly support the incorrect answer of user agent. See Appendix K for details.

Table 3 summarizes the results. Failure attribution varies across models and tasks. On web search, Qwen models fail more often on the user-agent side, while DeepSeek-V3.2, GPT-5, and Gemini-3 fail more often on the content-agent side. On web recommendation, neither side consistently dominates. On question answering, failures are mostly attributed to the user agent, but content-agent evidence still contributes non-negligible risk for DeepSeek-V3.2 and GPT-5. As one example, the question answering case study highlights a typical failure pattern. Even when content agents return the correct evidence, the user agent may still fail during answer synthesis and produce an answer at the wrong granularity (e.g., `The Sun` instead of `Light`). Together, these results suggest that improving Agentic Web systems requires progress on both user and content agents. Content agents need stronger retrieval, and user agents need better evidence interpretation and answer synthesis. More failure cases are provided in Appendix G.

## 6. Conclusion

We introduce AgentWebBench, the first comprehensive benchmark for evaluating agent performance in Agentic Web across four common web tasks. AgentWebBench has a decentralized architecture where a user agent coordinates with multiple content agents, each managing its own website domain. Comprehensive experiments across seven advanced large language models and three coordination strate-

gies show that decentralized access often underperforms centralized retrieval. However, this gap narrows as models scale and can reverse on the question answering task. Furthermore, AgentWebBench allows us to understand this paradigm beyond end-task accuracy. We provide insights that characterize ecosystem impact and identify improvement pathways. Specifically, we study traffic concentration, test-time scaling, interaction efficiency, and failure analysis. Overall, AgentWebBench provides a foundation for research in this emerging paradigm. Our results highlight next steps, including better coordination strategies, adaptive coordination that balances interaction cost and efficiency, and robust evidence retrieval and answer synthesis.

## Limitations

AgentWebBench is designed as a controlled testbed for studying coordination under decentralized access constraints. To achieve this, we deliberately fix the retrieval infrastructure and unified instruction template across all content agents: every website uses the same dense retrieval backend and returns results in a uniform format. This design parallels established benchmark methodology, where BEIR (Thakur et al., 2021) fixes the retrieval model to isolate cross-domain generalization and WebArena (Zhou et al., 2024) standardizes web environments to isolate agent planning. It enables fair comparison across coordination strategies without website design bias. Accordingly, all findings in this paper hold within this controlled context, and generalization to heterogeneous real-world settings remains an open question. This controlled setup does not capture several dimensions of real-world provider heterogeneity. In particular, our environment models (1) static rather than dynamically updated content, (2) a uniform rather than provider-specific retrieval backend, and (3) cooperative rather than adversarial content agents. Extending the benchmark to support heterogeneous provider-controlled systems with different APIs, policies, ranking objectives, authentication constraints, and latency profiles is an important direction for future work, which our modular decentralized design naturally supports.

## Impact Statement

This paper introduces AgentWebBench to help researchers evaluate decentralized information access in the Agentic Web. By providing a standardized and reproducible benchmark, it enables clear comparisons across models and coordination strategies, and it can help researchers build more effective user agents and content agents for the emerging Agentic Web. We also encourage careful use of this benchmark in deployment-focused settings. Decentralized nature of Agentic Web can shift attention toward a small set of sources, which may reduce information diversity. Moreover, this work may influence how agents are developed

and deployed on the web. As agents become more common in search and browsing, the research community shares a responsibility to support safe and responsible development through better evidence verification, transparent reporting, and safeguards for privacy.

## Acknowledgments

We thank Chen Xu, Cathy Jiao, Zichun Yu, Jingjie Ning, and Saahith Janapati for insightful discussions and feedback. This work is partially supported by Anaxi Labs.

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

# Contents of Appendix

# A. Additional Analysis on Web Traffic

In this section, we analyze how coordination strategies shape web traffic in AgentWebBench. We operationalize traffic as the frequency of cited sources, aggregated across queries. We focus on deep research, where answers must synthesize evidence from multiple websites. We first quantify content-agent visibility with GEO (Chen et al., 2025) (Table 4) as the additional analysis of Sec. 5.2. Then under a fixed Classic baseline, we compare how different cooperation strategies, including Multi-Agent, Tool$_E$, and Tool$_P$, affect web traffic (Fig. 5).

## A.1. Additional Analysis on Multi-Agent Cooperation

We analyze content agent visibility using the GEO metric proposed by Chen et al. (2025), which measures how evidence from multiple cited documents is distributed within a model-generated response. This analysis is conducted on the deep research task, where successful responses are expected to integrate information from multiple content agents rather than relying on a single dominant source. For each query, GEO assigns a normalized score to every cited document such that the scores sum to one, reflecting the relative contribution of each content agent. We compute GEO scores under three variants: word-based (Word), position-based (Pos), and their combination (Overall). Since absolute GEO scores are inherently normalized, we focus on their dispersion across cited agents. Specifically, for each response, we compute the standard deviation of GEO scores across all cited documents, and then average this statistic across queries with at least one citation. The average standard deviation quantifies the dispersion of evidence contributions across cited content agents.

Table 4 reports content agent visibility across different LLMs. Lower standard deviation indicates more balanced utilization of multiple content agents, while higher values suggest that evidence is concentrated on a small subset of agents. We additionally report the average number of cited content agents per response (#Cited Agents) and the percentage of responses without any citation (No-Ref.), which reflects complete visibility failure where no content agent is surfaced at all. In contrast to the global citation frequency in Sec. 5.2 (Fig. 2), which aggregates which sources are cited most often across queries, the visibility metrics here focus on the within-response distribution of citations—how evidence is apportioned among the content agents cited in each individual report.

From the table, Multi-Agent consistently cites fewer content agents per response than Classic across all models. In within-response dispersion (Word/Pos/Overall), Multi-Agent yields more balanced evidence distribution for Qwen 4B–80B but more concentrated distribution for DeepSeek-V3.2, GPT-5, and Gemini-3. Qwen3-4B under Multi-Agent exhibits severe visibility failure (31.12% No-Ref.), where nearly one-third of responses cite no content agent at all. These visibility patterns yield two takeaways that align with broader findings in the paper. First, for strong models (DeepSeek, GPT-5, Gemini-3), within-response concentration mirrors the global traffic concentration in Sec. 5.2: Multi-Agent's planning converges on fewer, high-utility agents both across queries and within each report, reinforcing the trade-off between

*Table 4.* Content agent visibility on the deep research task, measured by the standard deviation of GEO scores across cited content agents. No-Ref. (%) reports the percentage of responses without any citation. Word, Pos, and Overall correspond to word-based, position-based, and combined GEO variants, respectively (four decimal places to identify fine-grained differences).

| Model | Setting | #Cite | No-Ref. | Word ↓ | Pos. ↓ | Overall ↓ |
|---|---|---|---|---|---|---|
| Qwen3-4B | Classic | 2.93 | 0.60 | 0.1642 | 0.1569 | 0.1675 |
| | Multi-Agent | 2.63 | 31.12 | 0.0948 | 0.0931 | 0.0960 |
| Qwen3-14B | Classic | 2.97 | 1.81 | 0.1540 | 0.1443 | 0.1611 |
| | Multi-Agent | 2.67 | 1.51 | 0.0748 | 0.0766 | 0.0877 |
| Qwen3-30B-A3B | Classic | 3.35 | 2.42 | 0.1667 | 0.1589 | 0.1731 |
| | Multi-Agent | 2.53 | 0.60 | 0.1386 | 0.1315 | 0.1442 |
| Qwen3-80B-A3B | Classic | 4.11 | 2.11 | 0.1534 | 0.1496 | 0.1569 |
| | Multi-Agent | 2.55 | 0.91 | 0.1157 | 0.1125 | 0.1180 |
| DeepSeek-V3.2 | Classic | 6.59 | 8.16 | 0.1125 | 0.1147 | 0.1183 |
| | Multi-Agent | 3.64 | 0.60 | 0.1326 | 0.1340 | 0.1423 |
| GPT-5 | Classic | 6.74 | 24.17 | 0.0841 | 0.0846 | 0.0895 |
| | Multi-Agent | 5.13 | 3.32 | 0.1231 | 0.1192 | 0.1261 |
| Gemini-3 | Classic | 8.19 | 3.32 | 0.1146 | 0.1166 | 0.1221 |
| | Multi-Agent | 4.02 | 0.30 | 0.1456 | 0.1459 | 0.1500 |

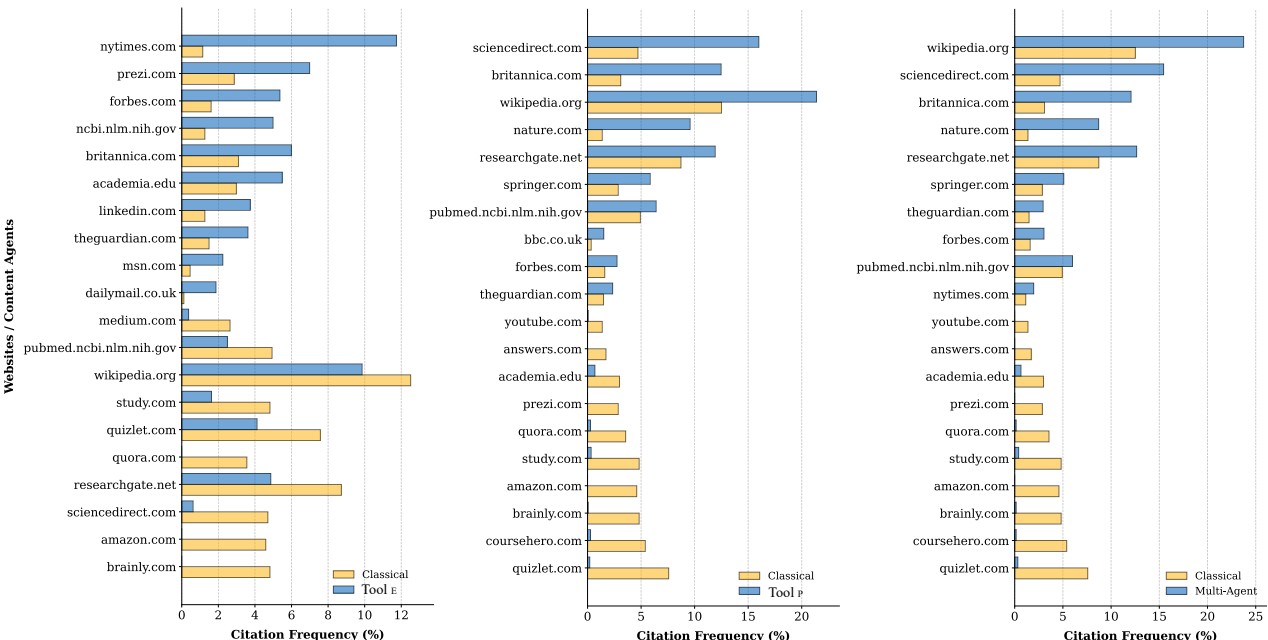

*Figure 5.* Impact of the different coordination strategies on web traffic. The charts illustrate the citation frequency of 20 most cited sources (websites or content agents) for Gemini-3 user agents on the deep research task. Each subplot is sorted vertically by the frequency difference (Tool$_E$ / Tool$_P$ / Multi-Agent minus Classical). The upper section highlights sources more frequently cited under the multi-agent coordination, while the lower section shows those preferred by Classical.

information reliability and source diversity. Second, for weaker models (e.g., Qwen3-4B), the 31% No-Ref. and the efficiency analysis in Appendix C—where limited #Agent and #Req. constrain coverage—together suggest that insufficient planning or coordination prevents content agents from being surfaced at all. This is consistent with the gains from test-time scaling (Sec. 5.3) and the coordination-strategy bottleneck discussed below: improving planning and coordination may help strong models retain utility while mitigating over-concentration, and help weak models overcome visibility failure.

### A.2. User Traffic for Tool-based Methods Tool$_P$ and Tool$_E$

Fig. 5 shows that even under the same base LLM (Gemini-3) and task setting (deep research), the coordination strategy substantially reshapes which sources become visible and thus cited. Since both Tool$_E$ and Tool$_P$ are tool-based (no content agents), the observed differences can be largely attributed to their website selection mechanisms rather than agent-to-agent interaction dynamics.

**Tool$_E$ (embedding selection) favors topical/professional outlets.** In the left subplot (Classical vs. Tool$_E$), Tool$_E$ increases citations to sources such as `nytimes.com`, `forbes.com`, `prezi.com`, and `ncbi.nlm.nih.gov`, while Classical more frequently cites general-purpose or study-aid platforms (e.g., `wikipedia.org`, `quizlet.com`, `brainly.com`, `amazon.com`). This pattern is consistent with embedding-based domain identification: selecting websites by nearest-neighbor similarity to the query tends to surface domains whose sampled in-domain documents are lexically/semantically close to the query, which can skew traffic toward news or professional publishers for many open-ended research prompts.

**Tool$_P$ (prompted selection) promotes authoritative knowledge bases and academic sources.** In the middle subplot (Classical vs. Tool$_P$), prompted selection yields a pronounced increase in citations to `wikipedia.org` and academic/curated sources such as `sciencedirect.com`, `britannica.com`, `researchgate.net`, and `nature.com`, while decreasing citations to homework/shortcut sites (e.g., `quizlet.com`, `coursehero.com`, `brainly.com`). We interpret this as an "authority prior" induced by LLM reasoning over website descriptions: when asked to select relevant domains for evidence synthesis, the model systematically prefers encyclopedic and scholarly repositories that are broadly applicable across topics.

**Relation to Multi-Agent traffic shifts.** The right subplot further amplifies the Tool$_P$ trend: multi-agent coordination concentrates citations on high-coverage knowledge bases (notably `wikipedia.org` at ∼25%) and consistently surfaces

scholarly platforms, while suppressing study-aid websites. Together, these results suggest that for deep research, moving from embedding-only selection (Tool$_E$) to reasoning-based selection (Tool$_P$) already shifts citations toward more authoritative sources, and full multi-agent coordination further strengthens this tendency via iterative refinement and adaptive exploration.

Table 2 further shows that Tool$_E$ is often the strongest overall on deep research task, while Multi-Agent is typically comparable to Tool$_P$. Together with Fig. 5, this suggests that a substantial portion of the gains on AgentWebBench can be achieved through better coordination strategy, whereas multi-agent coordination primarily reshapes how evidence is iteratively explored. This highlights coordination strategy as a key bottleneck, and motivates combining multi-agent interaction with more reliable coordination/verification to translate traffic shifts into consistent task-level improvements.

## B. Additional Results for Test Time Scaling

In the test time scaling analysis presented in Sec. 5.3, we compare LLMs that do not support thinking mode versus those that exclusively use thinking mode, specifically Qwen3-4B-Instruct versus Qwen3-4B-Thinking, and Qwen3-30B-A3B-Instruct versus Qwen3-30B-A3B-Thinking, focusing on web search and question answering tasks. In this section, we extend this analysis to all four tasks and additionally examine DeepSeek-V3.2, an LLM that supports dynamic switching between thinking and non-thinking modes, enabling a direct comparison of the same model architecture under different reasoning configurations. Table 5 presents comprehensive results for DeepSeek-V3.2 across all four tasks under both thinking and non-thinking modes. The comparison reveals several key patterns consistent with the findings in Sec. 5.3.

*Table 5.* Performance comparison across four tasks under different LLM-based methods. Tasks include web search, web recommendation, question answering (QA), and deep research. Metrics (%) are task-specific. "-" indicates not applicable. **Bold** and underline indicate the best and second-best results, respectively.

| LLM | Method | Web Search | | | | Web Recommendation | | | | QA | | Deep Research | | | |
|---|---|---|---|---|---|---|---|---|---|---|---|---|---|---|---|
| | | N@3 | N@5 | R@3 | R@5 | N@3 | N@5 | R@3 | R@5 | Acc. | F1 | KPR | KPC↓ | Clarity | Insight |
| | Classical | **49.37** | **50.66** | **57.63** | **60.73** | **1.07** | **1.07** | **1.07** | **1.07** | 15.09 | 12.40 | 65.26 | **1.36** | 88.73 | 80.15 |
| DeepSeek | Tool$_E$ | 32.99 | 34.63 | 37.01 | 40.96 | 0.36 | 0.36 | 0.36 | 0.36 | **16.98** | 8.57 | 66.17 | 1.91 | **90.18** | **85.71** |
| V3.2 | Tool$_P$ | 20.09 | 21.50 | 23.73 | 27.12 | 0.00 | 0.00 | 0.00 | 0.00 | 9.43 | 5.94 | 57.91 | 1.63 | 79.61 | 69.79 |
| (non-think) | Multi-Agent | 12.26 | 12.39 | 13.28 | 13.56 | 0.00 | 0.00 | 0.00 | 0.00 | 16.98 | **15.58** | 58.13 | 1.81 | 80.06 | 70.12 |
| | Classical | **49.02** | **49.37** | **55.37** | **56.21** | 0.40 | 0.40 | 0.71 | 0.71 | 24.53 | 19.51 | 66.54 | 1.27 | 87.16 | 80.48 |
| DeepSeek | Tool$_E$ | 33.98 | 35.05 | 37.85 | 40.40 | 0.00 | 0.00 | 0.00 | 0.00 | 20.75 | 17.05 | **68.23** | **1.11** | **88.61** | **83.56** |
| V3.2 | Tool$_P$ | 34.98 | 36.17 | 40.11 | 42.94 | **0.58** | **0.89** | 0.71 | **1.42** | 24.53 | 17.00 | 64.23 | 1.47 | 86.65 | 74.89 |
| (think) | Multi-Agent | 33.42 | 34.23 | 39.27 | 41.24 | 0.18 | 0.18 | 0.36 | 0.36 | **30.19** | **29.55** | 64.95 | 1.59 | 87.07 | 75.77 |

**Thinking mode significantly benefits multi-agent coordination.** For web search, Multi-Agent shows the most pronounced improvement under thinking mode, with N@3 increasing from 12.26% to 33.42%, nearly tripling performance. Tool$_P$ also improves substantially (20.09% to 34.98% N@3), while Classical performance remains stable (49.37% to 49.02% N@3) and Tool$_E$ shows only marginal gains (32.99% to 33.98% N@3). For question answering, Multi-Agent achieves the largest absolute improvement, with accuracy increasing from 16.98% to 30.19%, compared to Classical (15.09% to 24.53%) and Tool$_E$ (16.98% to 20.75%). For deep research, non-thinking mode already achieves strong performance across all methods (Clarity: 79.61%–90.18%, Insight: 69.79%–85.71%), and thinking mode maintains comparable or slightly improved performance. These gains, particularly for Multi-Agent on web search and question answering, align with the observation that explicit reasoning traces enable better information-seeking strategies and iterative coordination across multiple content agents.

**Method-specific patterns under thinking mode.** While Classical remains the strongest baseline across most tasks, thinking mode narrows the performance gap between Classical and agent-based methods. On web search, Tool$_P$ and Multi-Agent under thinking mode achieve performance closer to Classical (34.98% and 33.42% N@3, respectively) compared to non-thinking mode (20.09% and 12.26%). For deep research, both thinking and non-thinking modes achieve strong performance, with Tool$_E$ in non-thinking mode achieving the highest Clarity (90.18%) and Insight (85.71%) scores. This suggests that thinking mode particularly benefits methods that require complex coordination and iterative reasoning in tasks like web search and question answering, while deep research can achieve strong results in both modes.

**Task-dependent effectiveness.** Web recommendation shows limited improvement with thinking mode, with most methods achieving near-zero performance in both modes, consistent with the task's inherent difficulty noted in the main text. Deep research already achieves strong performance in non-thinking mode, suggesting that the task's requirements can be

*Table 6.* Mean interaction efficiency across tasks in AgentWebBench for Qwen3-4B and Gemini-3. Turn indicates the number of agent-generated responses; #Agent and #Req. denote the average number of contacted content agents and retrieval requests, respectively. Valid (%) reports the success rate of requests or returns.

| Task | User Agent | | | | Content Agent | |
|---|---|---|---|---|---|---|
| | Turn | #Agent | #Req. | Valid (%) | Turn | Valid (%) |
| **LLM: Qwen3-4B** | | | | | | |
| Web Search | 2.61 | 2.61 | 2.79 | 88.58 | 2.53 | 78.65 |
| Web Rec. | 7.63 | 6.02 | 9.88 | 84.88 | 2.69 | 53.99 |
| Question Answering | 3.38 | 1.36 | 2.06 | 91.37 | 3.83 | 47.62 |
| Deep Research | 4.61 | 3.60 | 4.05 | 96.20 | 2.24 | 52.54 |
| **LLM: Gemini-3** | | | | | | |
| Web Search | 3.29 | 4.64 | 6.26 | 99.10 | 7.98 | 56.42 |
| Web Rec. | 3.27 | 3.64 | 6.55 | 97.83 | 10.61 | 43.41 |
| Question Answering | 4.11 | 2.40 | 4.91 | 75.05 | 12.37 | 25.26 |
| Deep Research | 4.82 | 5.07 | 9.04 | 93.04 | 9.87 | 57.03 |

effectively met without explicit reasoning traces for DeepSeek-V3.2. The most pronounced benefits from thinking mode occur in web search and question answering, where multi-agent coordination particularly benefits from explicit reasoning to plan and execute information-seeking strategies.

These results complement the findings in Sec. 5.3 by demonstrating that the benefits of test time scaling extend beyond the Qwen series to models with flexible mode switching. The improvements are most pronounced for multi-agent coordination in web search and question answering tasks, while deep research already achieves strong performance in non-thinking mode, suggesting task-dependent effectiveness of explicit reasoning traces.

# C. Additional Results of Efficiency Analysis

Table 6 summarizes interaction efficiency metrics in AgentWebBench, including agent turns, the number of contacted content agents (#Agent), retrieval requests (#Req.), and validity rates (Valid %). Below are the detailed explanation of each statistic:

(a) **User agent: Turn.** The number of user agent responses generated before termination. Higher values indicate longer decision trajectories.

(b) **User agent: #Agent.** The number of distinct content agents contacted at least once. This measures breadth of exploration across websites.

(c) **User agent: #Req.** The total number of retrieval requests issued to content agents. Multiple requests to the same agent are counted separately. This measures the overall retrieval budget.

(d) **User agent: Valid (%).** The fraction of issued requests that are successfully parsed and executed by the environment. A request is invalid if it violates the tool schema, uses an unsupported target, or cannot be parsed into the required structured format. Low validity therefore indicates action-level errors, not retrieval quality.

(e) **Content agent: Turn.** The number of content agent responses returned for an instance. This is linked to how many retrieval requests are executed and how many agents are contacted.

(f) **Content agent: Valid (%).** The fraction of executed retrieval calls that return non-empty results. This captures whether a request retrieves any documents. It does not measure relevance. Low values can arise from poor query formulation, overly narrow constraints, or mismatched website selection.

Table 6 quantifies these trends using mean statistics. On web search, Gemini-3 contacts more agents and issues more requests than Qwen3-4B (#Agent 4.64 vs. 2.61, #Req. 6.26 vs. 2.79). It also achieves higher user-agent validity (99.10% vs. 88.58). At the same time, its content-agent validity is lower (56.42% vs. 78.65), and it requires more content-agent turns (7.98 vs. 2.53). On question answering, Gemini-3 again uses a larger retrieval budget (#Req. 4.91 vs. 2.06; #Agent 2.40 vs. 1.36), but it has lower user-agent validity (75.05% vs. 91.37) and lower content-agent validity (25.26% vs. 47.62). On deep research, Gemini-3 makes substantially more requests (#Req. 9.04 vs. 4.05) and contacts more agents (5.07 vs. 3.60), while maintaining high user-agent validity (93.04%). Web recommendation provides a counterexample. Qwen3-4B interacts more than Gemini-3 (Turn 7.63 vs. 3.27; #Agent 6.02 vs. 3.64; #Req. 9.88 vs. 6.55), yet this does not translate into strong task performance in Table 2. This indicates that interaction quantity alone is insufficient when intent inference and query

formulation are unstable.

From efficiency analysis, interaction efficiency is jointly determined by interaction budget and interaction effectiveness. Budget is captured by Turn, #Agent, and #Req. Effectiveness is proxied by validity on both sides. Strong performance typically requires both sufficient budget and sufficiently high validity. These results motivate adaptive budgeting and improved request formation to reduce unproductive interaction. Overall, efficiency in this type of Agentic Web architecture should not be equated with minimizing interaction counts alone. Instead, effective systems require an appropriate level of coordination, guided by accurate and deliberate planning, to balance interaction cost with information quality. As demonstrated in Table 2 and Fig. 3, base model capability and reasoning strength are foundational, but these results motivate future algorithmic innovations including adaptive interaction budgeting, planning-aware coordination strategies, and dynamic stopping criteria that can further optimize the efficiency-effectiveness tradeoff in this two-sided architecture.

## D. Implementation Details

**LLMs.** All LLMs used in our experiments are shown in Table 7. The maximum input length varies across models. For documents exceeding the maximum input length, we truncate the middle content of each document in the prompt using the same scale factor as in Huang et al. (2025d). We set the maximum output length to 8,192 tokens for fair comparison across all models.

Regarding thinking capabilities, as shown in Table 7:

- ×: Standard Instruct variants (e.g., Qwen1.5-14B, Qwen2.5-14B, Qwen3-*-Instruct-2507) do not natively support explicit thinking mode.
- ✓: Qwen3-Thinking variants (e.g., Qwen3-4B-Thinking-2507, Qwen3-30B-A3B-Thinking-2507, Qwen3-Next-80B-A3B-Thinking) are optimized exclusively for thinking mode, always performing step-by-step reasoning with improved quality and depth before generating the final answer. These models do not support non-thinking mode.
- ⇆: Certain Qwen3 models (e.g., Qwen3-14B), DeepSeek-V3.2, GPT-5 (including gpt-5-mini), and Gemini-3 (including gemini-3-flash-preview) support mode switching between the thinking mode (for complex logical reasoning, mathematical calculation, and coding tasks) and the non-thinking/fast mode (for efficient general-purpose dialogue). This flexible mode adjustment allows all these models to achieve optimal performance across diverse scenarios.

For models with ⇆ capability, we enable thinking mode, and set the reasoning effort of GPT-5 to medium.

*Table 7.* All LLMs used in experiments with maximum input and default output lengths, including thinking capability. The Thinking column indicates model capability: × = thinking mode not supported; ✓ = thinking mode only (forced step-by-step reasoning); ⇆ = supports switching/dynamic control between thinking and non-thinking/fast modes.

| Short Name | Full Name | Max Input Length | Max Output Length | Thinking |
|---|---|---|---|---|
| Qwen1.5-14B | Qwen/Qwen1.5-14B | 32,768 | 8,192 | × |
| Qwen2.5-14B | Qwen/Qwen2.5-14B | 131,072 | 8,192 | × |
| Qwen3-4B-Instruct | Qwen/Qwen3-4B-Instruct-2507 | 262,144 | 8,192 | × |
| Qwen3-4B | Qwen/Qwen3-4B-Thinking-2507 | 262,144 | 8,192 | ✓ |
| Qwen3-14B | Qwen/Qwen3-14B | 32,768 | 8,192 | ⇆ |
| Qwen3-30B-A3B-Instruct | Qwen/Qwen3-30B-A3B-Instruct-2507 | 262,144 | 8,192 | × |
| Qwen3-30B-A3B | Qwen/Qwen3-30B-A3B-Thinking-2507 | 262,144 | 8,192 | ✓ |
| Qwen3-80B-A3B | Qwen/Qwen3-Next-80B-A3B-Thinking | 262,144 | 8,192 | ✓ |
| DeepSeek-V3.2 | deepseek-ai/DeepSeek-V3.2 | 128,000 | 8,192 | ⇆ |
| GPT-5 | gpt-5-mini | 272,000 | 8,192 | ⇆ |
| Gemini-3 | gemini-3-flash-preview | 1,048,576 | 8,192 | ⇆ |

**Test Time Scaling.** When calculating the response length of models as shown in Fig. 3, we exclude summary responses, as the summary action always appears when the context is too long and bears limited relation to the model's performance.

## E. Impact of LLM Evolution

We investigate how the evolution of LLMs affects their performance in Agentic Web tasks. We select the Qwen series as the object of analysis and focus on 14B parameter models, as nearly all versions of Qwen are released in this size. Qwen-14B is excluded due to its short input length (2048 tokens), which limits performance in tasks requiring long context. Table 8

shows results across four tasks: web search, web recommendation, question answering, and deep research. The results indicate that newer Qwen versions achieve substantial improvements, especially in reasoning-intensive tasks. For example, QA accuracy increases from 0.00% (Qwen1.5) to 28.30% (Qwen3), and web search also rise sharply. These observations suggest that LLM development aligns with better performance in Agentic Web scenarios.

*Table 8.* Performance comparison across four tasks under different version LLMs. Tasks include web search, web recommendation, question answering (QA), and deep research. Metrics (%) are task-specific.

| LLM | Web Search | | | | Web Recommendation | | | | QA | | Deep Research | | | |
|---|---|---|---|---|---|---|---|---|---|---|---|---|---|---|
| | N@3 | N@5 | R@3 | R@5 | N@3 | N@5 | R@3 | R@5 | Acc. | F1 | KPR | KPC ↓ | Clarity | Insight |
| Qwen1.5-14B | 0.00 | 0.00 | 0.00 | 0.00 | 0.00 | 0.00 | 0.00 | 0.00 | 0.00 | 1.08 | 3.73 | 0.06 | 4.35 | 4.26 |
| Qwen2.5-14B | 0.00 | 0.00 | 0.00 | 0.00 | 0.00 | 0.00 | 0.00 | 0.00 | 5.66 | 4.01 | 7.32 | 0.18 | 6.74 | 6.89 |
| Qwen3-14B | 27.33 | 27.90 | 31.36 | 32.77 | 0.22 | 0.53 | 0.36 | 1.07 | 28.30 | 23.79 | 49.85 | 1.29 | 63.23 | 51.69 |

## F. Retrieval Backend Ablation

We adopt dense retrieval as the default backend for two reasons: it is consistent with the ClueWeb22 corpus and it represents the mainstream approach in modern RAG systems. To verify that our findings are not an artifact of this choice, we additionally evaluate two alternative retrieval backends on the web search task with Gemini-3:

- **BM25**: Keyword-based sparse retrieval, with 82,860,270 unique terms extracted from the 18,427,770 documents in our corpus.
- **Hybrid**: Reciprocal Rank Fusion (RRF) combining dense and BM25 retrieval.

Table 9 reports the results across all three backends. While dense retrieval yields the strongest absolute scores, our key findings remain consistent across all retrieval backends:

- $Tool_P$ is often better than $Tool_E$ (Sec. 5.1 (2)), which suggests that LLM reasoning helps pick more relevant websites than embedding similarity.
- Content agents are less stable on retrieval-heavy tasks (Sec. 5.1 (3)). $Tool_P$ and Multi-Agent mainly differ in whether they use content agents, and $Tool_P$ is usually stronger, which suggests the current content-agent retrieval strategy is not yet stable for retrieval-focused tasks.

Across all three retrieval backends, the relative strategy rankings remain consistent. This confirms that our conclusions are robust to the choice of retrieval backend.

## G. Case Study

This section provides qualitative case studies to complement the quantitative results in Table 2 and the failure analysis in Section 5.5. We present representative failure cases from the state-of-the-art model (Gemini-3) across three tasks: web search, web recommendation, and question answering. For each task, we analyze two cases (Case 1 and Case 2) that illustrate distinct failure modes at different stages of the multi-agent coordination pipeline, including user agent failures (e.g., intent misinterpretation, context tracking errors) and content agent failures (e.g., low retrieval recall, poor retrieval precision, evidence quality issues).

**Web Search.** On the web search task, agents must identify the correct website and retrieve the target document that matches the user's query. Failures typically occur at two levels: (a) the user agent may misinterpret query intent and route requests to irrelevant websites, or (b) the content agent may fail to retrieve the specific target document even when the correct website is selected. We present two representative cases illustrating these failure modes.

---

**Case 1: User Agent Failure on Web Search (Gemini-3)**

**[User Query]** dbl design

**[GT Answer]** {"website": "www.instagram.com", "doc title": "DBL Design (@dbldesign) • Instagram photos and videos"}

**[Content Agent "en.wikipedia.org"]** Design-Based Learning (DBL) is an inquiry-based educational pedagogy that

---

*Table 9.* Retrieval backend ablation on the web search task with Gemini-3. We compare dense (default), BM25, and hybrid (RRF) retrieval across all methods. Metrics (%) are reported as N@3, N@5, R@3, and R@5.

| Retrieval | Method | N@3 | N@5 | R@3 | R@5 |
|---|---|---|---|---|---|
| Dense (Ours) | Classic IR | 47.86 | 51.04 | 57.63 | 65.25 |
| | Classical | 52.21 | 53.77 | 60.17 | 63.84 |
| | Tool$_E$ | 39.56 | 40.49 | 44.63 | 46.89 |
| | Tool$_P$ | 47.52 | 49.07 | 56.50 | 60.17 |
| | Multi-Agent | 44.34 | 47.08 | 51.98 | 58.47 |
| BM25 | Classic IR | 28.12 | 31.48 | 33.33 | 41.53 |
| | Classical | 38.47 | 38.58 | 42.37 | 42.66 |
| | Tool$_E$ | 29.73 | 29.84 | 32.20 | 32.49 |
| | Tool$_P$ | 35.75 | 36.93 | 40.40 | 43.22 |
| | Multi-Agent | 33.07 | 34.93 | 38.98 | 43.50 |
| Hybrid | Classic IR | 43.44 | 47.41 | 51.98 | 61.86 |
| | Classical | 50.53 | 51.47 | 58.19 | 60.45 |
| | Tool$_E$ | 37.59 | 37.95 | 42.09 | 42.94 |
| | Tool$_P$ | 45.35 | 47.21 | 52.82 | 57.34 |
| | Multi-Agent | 44.94 | 46.35 | 53.11 | 56.50 |

integrates design thinking and problem-solving into the curriculum. Developed by Doreen Nelson in the 1980s, it emphasizes kinesthetic learning through an 8-step iterative process: investigating context, identifying needs, developing criteria, generating alternatives, selecting an alternative, prototyping/testing, producing, and evaluating. DBL focuses on the creation of artifacts to solve real-world problems, which has been shown to improve student engagement and performance in STEM subjects. It is closely related to instructional design frameworks like the ADDIE model and Universal Design for Learning (UDL), emphasizing flexibility and diverse engagement strategies.

**[Content Agent "github.com"]** While no specific repository exists for the complete design of a Diffusion Boundary Layer (DBL) for fuel cell ship hulls, several GitHub projects provide the computational and experimental foundations for such a design. Key resources include: 1) 'brli3/CFD' for implementing convection-diffusion and stagnation point flow models; 2) 'ECSIM/dbfc-dataset' for experimental fuel cell impedance and polarization data; and 3) 'CalebBell/fluids' for general fluid dynamics and friction factor calculations.

**[Content Agent "www.sciencedirect.com"]** On ScienceDirect, 'DBL design' primarily refers to several distinct concepts depending on the field: 1) Design-Based Learning (DBL), a pedagogical approach where students learn by designing solutions to problems; 2) Diffusion Boundary Layer (DBL), a concept in sensor design and electrochemistry regarding molecular transport near surfaces; 3) Multiple-Baseline Design, a research methodology in behavioral sciences; 4) Design-Build (DB), a construction project delivery method; and 5) Baseline Design, used in engineering for performance comparison.

**[Content Agent "stackoverflow.com"]** Search results on Stack Overflow for "Diffusion Boundary Layer (DBL) design" in the context of fuel cells and ship hulls indicate that the platform primarily hosts information on the numerical simulation of diffusion processes rather than physical engineering designs.

**[User Agent]**
1. clueweb22-en0028-59-01590: Design-based learning - Wikipedia
2. clueweb22-en0023-99-10424: Baseline Design - an overview — ScienceDirect Topics
3. clueweb22-en0041-18-02735: python - Finite difference method for 3D diffusion/heat equation - Stack Overflow
4. clueweb22-en0021-68-08764: python - Applying neumann boundary conditions to the diffusion equation - Stack Overflow
5. clueweb22-en0033-07-11750: GitHub - ECSIM/dbfc-dataset: Single DBFC Dataset

**Search Requests of User Agent:**
R1: search({"query": "dbl design", "websites": ["en.wikipedia.org", "www.researchgate.net", "github.com", "www.sciencedirect.com"]})
R2: search({"query": "Design-Based Learning (DBL) educational design", "websites": ["en.wikipedia.org", "www.researchgate.net", "www.britannica.com", "www.sciencedirect.com"]})
R3: search({"query": "D̈iffusion Boundary LayerD̈BL design OR D̈ouble Bottom LineD̈BL design", "websites": ["en.wikipedia.org", "www.sciencedirect.com", "www.researchgate.net"]})
R4: search({"query": "Diffusion Boundary Layer DBL design fuel cell ship hull", "websites": ["www.sciencedirect.com", "github.com", "stackoverflow.com"]})

---

**Analysis:** The user agent incorrectly interpreted the query intent and routed the request to irrelevant content agents. It treated "dbl design" as an academic or technical acronym query and expanded "DBL" into multiple conceptual meanings, restricting the search to scholarly and developer-oriented websites. This led to concept-level results with high semantic variance and no strong entity match. As a consequence, the agent failed to consider a navigational or brand-oriented interpretation and excluded social platforms such as Instagram, where the ground-truth entity is located.
**Failure Mode:** Intent misinterpretation. The agent incorrectly classified the query as an academic/technical acronym search rather than a navigational/brand search, leading to wrong website selection.

---

**Case 2: Content Agent Failure on Web Search (Gemini-3)**

**[User Query]** uninstall trend micro

**[GT Answer]** {"website": "answers.microsoft.com", "doc title": "How can I completely remove Trend Micro from my computer? - Microsoft Community"}

---

**Search Requests of User Agent:**
R1: search({"query": "how to uninstall trend micro", "websites": ["www.wikihow.com", "answers.microsoft.com", "www.reddit.com", "community.spiceworks.com"]})
R2: search({"query": "how to uninstall trend micro security windows mac", "websites": ["www.wikihow.com", "www.answers.com", "www.quora.com", "www.youtube.com"]})

---

**Content Agent "answers.microsoft.com"**
**[Received Query]** how to uninstall trend micro
**[Search Requests]**
R1: search("how to uninstall trend micro site:answers.microsoft.com")
Retrieved Documents:
1. clueweb22-en0035-37-06613: Uninstall Trend Micro OfficeScan?
2. clueweb22-en0012-70-09332: Why can't I uninstall/remove Trend Micro completely from my
3. clueweb22-en0038-61-18799: Uninstalling Trend Micro remnants from Windows 10

**[Return Documents]** clueweb22-en0035-37-06613, clueweb22-en0038-61-18799, clueweb22-en0012-70-09332

---

**Analysis:** The user agent correctly routed the query to the ground-truth website (answers.microsoft.com), but the content agent failed to retrieve the specific Microsoft Community thread needed for the answer. With a single short site-restricted query, it returned only loosely related discussions (e.g., OfficeScan uninstall or leftover-removal posts), indicating low recall and weak intent matching for "completely remove Trend Micro". As a result, the pipeline misses the ground-truth evidence despite correct website selection.
**Failure Mode:** Low retrieval recall. The content agent's site search query was too generic and failed to match the specific target document, resulting in only tangentially related results.

---

These two cases demonstrate that web search failures can stem from either incorrect website selection (Case 1) or insufficient retrieval precision within the correct website (Case 2). The first case shows how intent misinterpretation leads to routing errors, while the second illustrates that even correct website selection does not guarantee success if the content agent's

retrieval is too generic.

**Web recommendation.** In the web recommendation task, agents must infer the user's next browsing intent from their browsing history and recommend the most relevant next document. This requires tracking salient entities and evolving context across the history. Failures often occur when the agent loses track of key entities or drifts to incorrect topic interpretations. We present two cases: one where the user agent fails to track context, and another where the content agent fails to retrieve the target document despite correct website selection.

---

**Case 1: User Agent Failure on Web Recommendation (Gemini-3)**

**[User Browsing History]**
1. Cassidy/Quotes - Overwatch Wiki
2. LEGO Lane Stadium — The Key Play
3. The Peanuts Movie - YouTube
4. YOU Trailer (2018) Netflix - YouTube
5. 'Dateline' examines murder of Maurizio Gucci — NBC4 WCMH-TV
6. YouTube
7. DeSantis Covid-19 mystery- POLITICO
8. The 15 Most Annoying Celebrity Couples That Get On Our Nerves
9. xQc playlist 2021 - YouTube
10. The Dark Side of K-Pop: Assault, Prostitution, Suicide, and Spycams : popheads

**[Next Document]**
{"website": "twitter.com", "doc title": "xQcOWUpdates on Twitter: BREAKING: xQc has just recieved a DMCA strike for a clip from May of 2019. This is very bad for his channel and it could be deleted at any point with no warning if they find more violations. / Twitter"}

**[User Agent]**
1. clueweb22-en0020-00-11267: 10 Best Roblox Games in 2021 - News+ on RCTI+
2. clueweb22-en0009-84-18751: Chi Chi London — Women's Clothing & Fashion Dresses US – Chi Chi London US
3. clueweb22-en0026-47-19064: The Dark Side of K-Pop: Assault, Prostitution, Suicide, and Spycams : popheads
4. clueweb22-en0042-33-06382: NotSo Superficial - YouTube
5. clueweb22-en0028-22-06863: BANNED K-POP MUSIC VIDEOS - TOO HOT FOR TV! [Part 1] - YouTube
6. clueweb22-en0044-34-00412: Top 10 Biggest Corporate Scandals - YouTube

---

**Search Requests of User Agent:**
`R1:` search({"query": "major music industry scandals and k-pop controversies explained", "websites": ["en.wikipedia.org", "www.youtube.com", "www.reddit.com"]})

---

**Analysis:** The user agent fails to track the salient entity and evolving context in the browsing history. Although the history contains multiple xQc-related YouTube entries, the agent instead interprets the session as "music-industry scandals / K-pop controversies" and issues a broad topical query restricted to Wikipedia/YouTube/Reddit. This causes a mismatch with the ground-truth next page, which is a timely Twitter update about xQc's DMCA strike.
**Failure Mode:** Context tracking failure. The agent lost track of the salient entity (xQc) in the browsing history and drifted to a different topic interpretation, while also missing the appropriate source (twitter.com) for real-time updates.

---

**Case 2: Content Agent Failure on Web Recommendation (Gemini-3)**

**[User Browsing History]**
1. Ana de Armas Measurements: Height, Weight, Bra Size, Breast Size, & More
2. Kodak Black Height, Weight, Age, Girlfriend, Family & Biography
3. Bernadette Peters Height, Weight, Measurements, Bra Size, Shoe, Biography
4. Song Ji-a Height, Weight, Net Worth, Age, Birthday, Wikipedia, Who, Nationality, Biography — TG Time
5. Taylor Russell — Body Measurements

6. Bella Hadid Height in cm, Meter, Feet and Inches, Age, Bio

7. How tall is Zendaya? Real Height Revealed! - YouTube

8. Tom Holland Height in cm, Meter, Feet and Inches, Age, Bio

**[Next Document]**

{"website": "www.quora.com", "doc title": "Is Zendaya actually 510? - Quora"}

---

**Search Requests of User Agent:**

R1: search({"query": "Zendaya and Tom Holland height difference comparison", "websites": ["en.wikipedia.org", "www.youtube.com", "www.reddit.com", "www.imdb.com", "www.quora.com"]})

R2: search({"query": "celebrity height comparison chart with photos", "websites": ["en.wikipedia.org", "www.youtube.com", "www.reddit.com", "www.imdb.com", "www.quora.com"]})

---

**Content Agent "www.quora.com"**

**[Received Query]** Zendaya and Tom Holland height difference comparison

**[Search Requests]**

R1: search("Zendaya and Tom Holland height difference site:quora.com")

Retrieved Documents:

1. clueweb22-en0002-18-14059: Is Tom Holland dating his Spider-Man co-star Zendaya? - Quora

2. clueweb22-en0008-35-08714: Is Zendaya actually 5'10? - Quora

2. clueweb22-en0016-65-08451: Is Zendaya cheating on Tom Holland with Timoth Chalamet? - Quora

R2: search("Tom Holland height site:quora.com")

Retrieved Documents:

1. clueweb22-en0002-18-14059: Is Tom Holland dating his Spider-Man co-star Zendaya? - Quora

2. clueweb22-en0008-35-08714: Is Zendaya actually 5'10? - Quora

3. clueweb22-en0016-65-08451: Is Zendaya cheating on Tom Holland with Timoth̆00e9e Chalamet? - Quora

R3: search("How tall is Tom Holland? site:quora.com")

Retrieved Documents:

1. clueweb22-en0002-18-14059: Is Tom Holland dating his Spider-Man co-star Zendaya? - Quora

2. clueweb22-en0008-35-08714: Is Zendaya actually 5'10? - Quora

3. clueweb22-en0016-65-08451: Is Zendaya cheating on Tom Holland with Timoth̆00e9e Chalamet? - Quora

R4: search("What is Tom Holland's height according to Quora? site:quora.com")

Retrieved Documents:

1. clueweb22-en0002-18-14059: Is Tom Holland dating his Spider-Man co-star Zendaya? - Quora

2. clueweb22-en0008-35-08714: Is Zendaya actually 5'10? - Quora

3. clueweb22-en0016-65-08451: Is Zendaya cheating on Tom Holland with Timoth̆00e9e Chalamet? - Quora

R5: search("What is Tom Holland's height? site:quora.com")

Retrieved Documents:

1. clueweb22-en0002-18-14059: Is Tom Holland dating his Spider-Man co-star Zendaya? - Quora

2. clueweb22-en0008-35-08714: Is Zendaya actually 5'10? - Quora

3. clueweb22-en0016-65-08451: Is Zendaya cheating on Tom Holland with Timoth̆00e9e Chalamet? - Quora

R6: search("What is Tom Holland's exact height site:quora.com")

Retrieved Documents:

1. clueweb22-en0002-18-14059: Is Tom Holland dating his Spider-Man co-star Zendaya? - Quora

2. clueweb22-en0008-35-08714: Is Zendaya actually 5'10? - Quora

3. clueweb22-en0016-65-08451: Is Zendaya cheating on Tom Holland with Timoth̆00e9e Chalamet? - Quora

R7: search("How tall is Tom Holland according to Quora?")

Retrieved Documents:

1. clueweb22-en0002-18-14059: Is Tom Holland dating his Spider-Man co-star Zendaya? - Quora

2. clueweb22-en0008-35-08714: Is Zendaya actually 5'10? - Quora

---

3. clueweb22-en0016-65-08451: Is Zendaya cheating on Tom Holland with Timoth Chalamet? - Quora

`R8`: search("site:www.quora.com Tom Holland height")
`End`

---

**Analysis:** The user agent correctly selects the ground-truth domain (quora.com) and sends a relevant query about "Zendaya and Tom Holland" height comparison, but the content agent's site search exhibits poor intent matching and unstable ranking. Across many redundant reformulations, it repeatedly surfaces tangential relationship/gossip threads (e.g., dating/cheating) and fails to reliably retrieve the exact target thread ("Is Zendaya actually 5'10/510?"). This retrieval noise and weak disambiguation prevent the pipeline from recommending the ground-truth next document despite correct website selection.

**Failure Mode:** Poor retrieval precision. The content agent's site search returns noisy, tangentially related results instead of the exact target document, indicating weak query-document matching and ranking instability.

---

These cases highlight two distinct failure patterns on web recommendation: context tracking failures where the agent loses track of salient entities (Case 1), and retrieval precision failures where the content agent cannot reliably retrieve the target document despite correct website selection (Case 2). Both demonstrate the challenges of maintaining context coherence and achieving precise document retrieval in recommendation scenarios.

**Question answering.** On the question answering task, agents must retrieve relevant evidence from multiple sources and synthesize accurate answers. Failures can occur at the retrieval stage (content agents return conflicting or incomplete evidence) or at the synthesis stage (user agent fails to correctly interpret and combine evidence). We present two cases: one demonstrating a synthesis error despite correct evidence retrieval, and another showing how conflicting evidence from content agents leads to incorrect answers.

---

**Case 1: User Agent Failure on Question Answering (Gemini-3)**

**[User Query]** In Plato's analogy of the sun, what element does he compare to the Form of the Good in terms of its role in enabling knowledge?

**[GT Answer]** Light

**[Content Agent "en.wikipedia.org"]** In Plato's *Republic*, the analogy of the sun compares the Sun in the visible realm to the Form of the Good in the intelligible realm. The Sun provides light, which allows the eye to see physical objects, while the Form of the Good provides truth and reality, allowing the soul to understand the Forms. Just as the Sun is responsible for the growth and life of visible things without being growth itself, the Form of the Good is the source of the existence and essence of all other Forms, yet it transcends 'being' in rank and power.

**[Content Agent "www.britannica.com"]** In Plato's 'Republic', the analogy of the sun illustrates the Form of the Good's role in the intelligible world. Just as the sun provides light that allows the eyes to see and objects to be visible in the physical world, the Form of the Good provides truth and reality, enabling the soul to know and the Forms to be intelligible. Furthermore, the sun is the cause of growth and generation in the visible world, while the Form of the Good is the source of the being and essence of the Forms in the intelligible world.

**[User Agent]** The Sun ×

---

**Search Requests of User Agent:**
`R1`: search({"query": "Plato's analogy of the sun Form of the Good comparison", "websites": ["en.wikipedia.org", "www.britannica.com"]})

---

**Analysis:** The user agent correctly retrieves relevant evidence from both content agents, which explicitly state that "the Sun provides light" and that light enables vision. However, it fails to synthesize the answer correctly: it confuses the two levels of the analogy and selects "The Sun" (the visible-world entity) instead of "Light" (the specific element that enables knowledge/vision). The query asks for the element that plays the enabling role, not the entity being compared.

**Failure Mode:** Answer synthesis error. Despite having correct evidence, the agent misidentifies the answer granularity

---

by selecting the wrong level of the analogy (the entity being compared rather than the enabling element).

---

**Case 2: Content Agent Failure on Question Answering (Gemini-3)**

[**User Query**] Between the 7th and 12th centuries, if the Norse had never invaded and interacted with the Anglo-Saxons, what would have been the Modern English word for "sister"?

[**GT Answer**] Swister

[**Content Agent "en.wikipedia.org"**] The word 'sister' originates from the Proto-Indo-European root *swésōr, which evolved into Proto-Germanic *swestēr and subsequently Old English 'sweostor'. Without the influence of Old Norse 'systir', the native Old English 'sweostor' would have naturally developed into Middle English 'swester' (or 'suster' in some dialects) as the 'eo' diphthong simplified to 'e'. This would likely have resulted in a modern English form such as 'swester'. The modern spelling and pronunciation with 'i' (sister) are attributed to the heavy influence of the Old Norse cognate during the Viking Age, which displaced the native vowel.

[**Content Agent "www.britannica.com"**] The word "sister" derives from the Proto-Indo-European root *swésōr*, which became *swestēr* in Proto-Germanic and *sweostor* in Old English. The Modern English form "sister" was heavily influenced by the Old Norse cognate *systir*, which displaced the native Old English development. Without this Norse influence, *sweostor* would have likely evolved into "swester" or "swister" in Modern English, following standard phonological shifts (where the Old English diphthong *eo* typically simplified to *e*), similar to the evolution of "heaven" from *heofon*.

[**Content Agent "www.merriam-webster.com"**] The word 'sister' is derived from a blend of the Old English 'sweostor' and the Old Norse 'systir'. Without the influence of Old Norse, the native Old English development of 'sweostor' resulted in the Middle English form 'suster'. The modern spelling and pronunciation with an 'i' are specifically attributed to the Norse influence; the native OE form would have likely persisted as 'suster' (akin to how 'brother' and 'mother' developed) rather than 'sister'. Both forms share a common ancestor in Proto-Germanic *swestēr, and are cognate with Latin 'soror' and Sanskrit 'svasṛ'.

[**Content Agent "www.merriam-webster.com"**] A hypothetical modern English word for sister without Old Norse influence would likely be **suster** (pronounced to rhyme with 'buster'). The modern word 'sister' is a blend of Middle English *suster* (from Old English *sweostor*) and *sister* (from Old Norse *systir*). Without the Norse influence on the vowel, the Old English *sweostor* would have naturally evolved into the Middle English *suster*, resulting in the modern form 'suster'.

[**User Agent**] suster ✕

---

**Search Requests of User Agent:**
`R1`: search({"query": "etymology of sister and Old English sweostor development without Norse influence", "websites": ["en.wikipedia.org", "www.merriam-webster.com", "www.britannica.com"]})
`R2`: search({"query": "hypothetical modern english word for sister without norse influence", "websites": ["en.wikipedia.org", "www.etymonline.com", "www.merriam-webster.com"]})

---

**Analysis:** The content agents return conflicting and incomplete evidence. Wikipedia suggests "swester", Britannica mentions both "swester" and "swister", while Merriam-Webster consistently proposes "suster". None of these sources clearly identifies "swister" as the definitive answer, and the evidence variance confuses the user agent, which selects "suster" based on the most frequently mentioned variant.
**Failure Mode:** Evidence quality failure. The content agents retrieve documents with conflicting and imprecise information, preventing the user agent from synthesizing the correct answer despite correct website selection and query formulation.

---

These cases illustrate two critical failure modes on question answering: synthesis errors where the agent misinterprets correct evidence (Case 1), and evidence quality failures where conflicting information from multiple sources prevents accurate

synthesis (Case 2). The first case shows that having correct evidence does not guarantee correct answers if the agent fails to synthesize at the right granularity, while the second demonstrates how inconsistent evidence across sources can mislead the agent.

## H. Performance Comparison of Tool$_P$ and Multi-Agent

Table 2 reveals task-dependent performance patterns between Tool$_P$ and Multi-Agent. While both leverage LLM reasoning for website selection, they differ fundamentally: Tool$_P$ uses direct tool-based retrieval, whereas Multi-Agent enables agent-to-agent communication with autonomous reasoning and iterative interactions. For web search, Tool$_P$ consistently outperforms Multi-Agent (e.g., Gemini-3: N@3 47.52 vs. 44.34). This reflects Tool$_P$'s deterministic access to ranked documents versus Multi-Agent's iterative exploration, which can introduce noise. However, as shown in Sec. 5.3 and Sec. 5.4, improved coordination strategies could bridge this gap. For question answering, Multi-Agent matches or slightly exceeds Tool$_P$, particularly for stronger models (Gemini-3: identical accuracy 67.92%, but higher F1 61.10 vs. 60.26). This demonstrates Multi-Agent's strength in adaptive query refinement and evidence integration through iterative content agent interactions. For deep research, both methods show comparable results. These comparisons indicate that performance differences stem from task characteristics and coordination limitations rather than architectural weaknesses. Our analysis suggests that improved coordination strategies, test time scaling, and efficiency optimizations will enable Multi-Agent to excel in the future Agentic Web ecosystem.

## I. Near-zero Performance on Web Recommendation

The web recommendation task exhibits consistently near-zero performance across all experimental settings in our evaluation, leading us to exclude it from detailed analysis in Table 2. This outcome reflects the intrinsic difficulty of the task. Specifically, web recommendation requires models to infer latent and often evolving user intent from historical browsing behaviors, then generate effective queries to retrieve web documents that users are likely to click next. User intent in this context is highly contextual, ambiguous, and subject to rapid change, making the task inherently challenging. This example below demonstrates the challenge of intent inference: the browsing history spans diverse domains (medical information, gaming, music, legal records, entertainment), yet the next webpage is a technical GitHub issue about a systemd plugin. There is no obvious semantic connection between the historical browsing patterns and the target page, requiring models to capture subtle, potentially non-linear relationships or external context that may not be evident from the browsing sequence alone. Such cases are common in real-world web recommendation scenarios, where user behavior can shift rapidly between unrelated topics based on external factors, current tasks, or serendipitous discovery.

---

<User Browsing History>
1. Cold sores (fever blisters): Causes, symptoms, treatment, and more
2. Tongue bumps: Causes, when to see a doctor, and treatment
3. Canker Sores — Shoppers Drug Mart®
4. Flight Rising — Flight Rising Wiki — Fandom
5. FAQ: What are FIPS codes?
6. Volunteer Guardian Program
7. Lie Bumps (Transient Lingual Papillitis)
8. Flintstones Vitamins SHOCKING Reviews 2021 - Does It Really Work?
9. Want to learn Mandoa? Hopefully I can help - Learn Mandoa
10. twenty one pilots (@twentyonepilots) / Twitter
11. Illinois Arrests – Find arrest records in Illinois
12. Using Guides - Dozuki
13. Clerks — Tennessee Administrative Office of the Courts
14. All Products — Smithfield
15. 50 Best F1 Memes - Merry Memes
16. Show More button in descriptions has disappeared! - YouTube Community
17. TWIN vs TWIN REAL TELEPATHY TEST - YouTube

<Next Webpage>
Input plugin systemd cannot be loaded · Issue #1696 · fluent/fluent-bit · GitHub

---

We evaluate QueryGen (He et al., 2025), a query-generation baseline that maps user browsing history to retrieval queries, in our web recommendation setting (Table 10). The resulting near-zero N@k and R@k scores are consistent with the broader

*Table 10.* QueryGen performance (%) on our web recommendation setting. QueryGen is a non-agent baseline.

| Model | N@3 | N@5 | R@3 | R@5 |
|---|---|---|---|---|
| Qwen3-4B-QueryGen | 0.00 | 0.00 | 0.00 | 0.00 |
| Qwen3-14B-QueryGen | 0.76 | 0.76 | 1.07 | 1.07 |
| Qwen3-30B-A3B-QueryGen | 0.18 | 0.32 | 0.36 | 0.71 |
| Qwen3-80B-A3B-QueryGen | 0.89 | 0.89 | 1.42 | 1.42 |
| DeepSeek-V3.2-QueryGen | 0.76 | 0.90 | 1.07 | 1.42 |
| GPT-5-QueryGen | 0.71 | 1.02 | 0.71 | 1.42 |
| Gemini-3-QueryGen | 1.21 | 1.21 | 1.78 | 1.78 |

*Table 11.* Zero-shot ClueWeb-Reco benchmarking test results (%) on candidate item ranking (He et al., 2025).

| Model | Recall@10 | NDCG@10 | Recall@50 | NDCG@50 | Recall@100 | NDCG@100 |
|---|---|---|---|---|---|---|
| GPT-3.5-Turbo-QueryGen | 0.88 | 0.58 | 1.56 | 0.73 | 2.54 | 0.89 |
| GPT-4o-QueryGen | 0.68 | 0.27 | 1.76 | 0.50 | 3.12 | 0.72 |
| Gemini-2.5-Flash-QueryGen | 0.68 | 0.42 | 1.46 | 0.58 | 2.64 | 0.77 |
| GPT-4.1-QueryGen | 1.07 | 0.50 | 1.95 | 0.68 | 2.54 | 0.77 |
| Claude-Sonnet-4-QueryGen | 0.68 | 0.32 | 1.66 | 0.52 | 2.15 | 0.60 |
| DeepSeek-V3-QueryGen | 1.27 | 0.82 | 2.64 | 1.11 | 3.71 | 1.29 |
| Kimi-K2-QueryGen | 0.39 | 0.22 | 1.56 | 0.50 | 2.34 | 0.62 |
| Llama-4-Maverick-QueryGen | 0.29 | 0.15 | 0.88 | 0.28 | 2.05 | 0.47 |
| Qwen3-235B-QueryGen | 0.88 | 0.46 | 2.34 | 0.77 | 3.03 | 0.88 |

pattern reported in Table 2, and align with prior large-scale evaluations: on the ORBIT benchmark (He et al., 2025), even state-of-the-art LLMs achieve extremely low candidate-ranking performance, with the best NDCG@50 remaining below 1% (Table 11). Together, these results indicate that the low absolute numbers are not anomalous, but rather reflect the intrinsic difficulty of intent inference and proactive next-page prediction from noisy, rapidly shifting user behavior.

Importantly, this difficulty also makes web recommendation a meaningful stress test for Agentic Web systems. A core capability of web agents is to infer, track, and act on latent user intent over time; the uniformly poor results therefore highlight a key limitation of current agent architectures, especially for long-horizon intent modeling and proactive information seeking, and point to clear directions for future work.

# J. Details of Four Tasks in AgentWebBench

AgentWebBench supports a diverse set of tasks reflecting common information-seeking behaviors in agentic information ecosystems while requiring different forms of multi-agent coordination. We describe each task in terms of its motivation, interaction structure, dataset construction, and evaluation protocol.

## J.1. Web Search

The web search task evaluates a user agent's ability to identify and rank relevant documents under access constraints. Given a query, the user agent must return a ranked list of document identifiers. Unlike traditional retrieval benchmarks with open access, the user agent can only obtain candidate documents through content agents, each restricted to a single website.

We adapt 354 queries from the MS MARCO Web Search test set, mapped to the sandbox corpus to preserve relevance. Evaluation uses standard information retrieval metrics, including NDCG (N@k) and Recall (R@k). This task emphasizes precise content localization and enables credit assignment analysis, such as whether the user agent selects the correct content agents and whether content agents successfully retrieve ground-truth documents.

## J.2. Web Recommendation

The web recommendation task models personalized information access, where the coordinator must infer a user's next intent based on browsing history and recommend relevant web documents. The user agent interprets the observed interaction sequence, formulates an implicit query representing the anticipated intent, and coordinates with content agents to retrieve candidate documents.

This task is constructed using 281 user histories adapted from the ORBIT benchmark, which provides realistic browsing sequences and relevance judgments. Only instances whose target documents appear in the AgentWebBench corpus are retained. Evaluation is conducted using ranking-based metrics, including NDCG (N@k) and Recall (R@k). This task emphasizes user intent inference under partial information in access-controlled agentic information ecosystems.

### J.3. Question Answering

The question answering task focuses on producing concise factual responses while allowing multi-step search and reasoning. Many queries require aggregating evidence from multiple closed-domain sources, especially when no single content agent contains a complete answer. The user agent must decide which content agents to consult and progressively synthesize the final answer through repeated interactions.

We derive 53 queries from the short-answer subset of DeepResearchGym and filter them to ensure corpus coverage. The user agent may issue multiple retrieval requests before generating an answer, but must determine when sufficient evidence has been collected. Performance is evaluated for correctness using LLM-as-the-judge accuracy and token-level F1. This task emphasizes correctness and enables analysis of unnecessary agent invocations and failure attribution.

### J.4. Deep Research

The deep research task models complex information-seeking scenarios in which users request comprehensive and structured reports on open-ended topics. Unlike short-form question answering, this task requires iterative planning, multi-step retrieval, and synthesis of evidence across multiple sources. The user agent must decompose the query, decide which content agents to consult at different stages, and progressively refine the report through repeated interaction.

We adapt 331 queries from DeepResearchGym (Coelho et al., 2025) whose supporting documents are available within the AgentWebBench sandbox, ensuring that performance depends on agent interaction rather than external knowledge. Evaluation follows the DeepResearchGym protocol, assessing both report relevance and quality. Relevance is measured using key point recall (KPR), which evaluates whether salient points extracted from ground-truth documents are covered by the generated report, and key point contradiction (KPC), which penalizes factual inconsistencies. Report quality is further evaluated using an LLM-as-a-Judge framework along Clarity and Insight dimensions. This task emphasizes content quality and is particularly suited for analyzing content agent visibility and information integration in multi-agent agentic information ecosystems.

## K. Judgment Protocol for Failure Mode Attribution

In the two-sided Agentic Web architecture, failures can originate from different stages of coordination. The user agent may misplan or select inappropriate content agents, while content agents may retrieve incomplete, irrelevant, or misleading evidence that propagates to incorrect outputs. We attribute each incorrect prediction exclusively to either the user agent or the content agents on tasks with exact ground truth as shown in Table 3.

### K.1. Analytical Approaches for Different Task Types

We employ different analytical approaches for different task types. For web search and web recommendation, which have precise ground-truth documents, we perform deterministic failure attribution: we definitively assign responsibility to either the user agent or content agents based on whether ground-truth websites were contacted. For question answering, which is a generative task, we analyze failures through an evidence-risk lens: we assess whether incorrect answers are induced by misleading content-agent evidence, rather than performing deterministic attribution. This reflects the different nature of these tasks: discriminative tasks (web search, web recommendation) allow clear binary attribution based on ground-truth contact, while generative QA requires evaluating the risk that retrieved evidence misleads the user agent. Deep research is excluded from both attribution and risk analysis because it lacks absolute ground-truth answers and emphasizes open-ended synthesis, where multiple valid responses may exist. Instead, we analyze deep research through content agent visibility metrics (Appendix A), which characterize how effectively models integrate information from multiple sources.

### K.2. Attribution Rules for Web Search and Web Recommendation

For web search and web recommendation, we employ rule-based attribution. The decision rules are as follows:

- **Web search.** If the user agent does not contact any content agent whose website hosts ground-truth relevant documents, the failure is attributed to the user agent. Otherwise, it is attributed to the content agents (site-level retrieval failing to surface ground-truth documents within the allowed returns).
- **Web recommendation.** The same rule as web search: if the user agent never contacts a content agent hosting the target document, attribute to the user agent; otherwise attribute to the content agents.

### K.3. LLM-as-Judge for Question Answering

For question answering, we employ an LLM-as-Judge procedure using GPT-5-nano (OpenAI, 2025b) to determine the primary cause of an incorrect answer, considering both efficiency and accuracy. The judge receives: the user question, the user agent's final wrong answer, and the retrieved evidence returned by content agents. The labeling instruction is: attribute to content agents if the incorrect answer is directly grounded in retrieved evidence that is incorrect, irrelevant, or misleading; otherwise attribute to the user agent when the error arises from deficient search planning, poor content agent selection, or faulty synthesis despite adequate evidence being available.

We select GPT-5-nano as the judger as it is a state-of-the-art LLM that ensures reliability. Through multiple runs, we find that the evaluation results are quite stable. To further enhance evaluation reliability, Table 3 shows that we conduct each judgment three times and take the majority vote as the final attribution. The instruction template for question answering judge is as follows:

---

You are an impartial judge evaluating evidence grounding for Question Answering.

Task:
Determine whether the provided Content provides substantive and logical support for the given Answer with respect to the Question.

Definition of "related":
- The Content must contain factual information that directly supports, explains, or verifies the key constraints and reasoning required by the Question in order to justify the Answer.
- Mere mention of the Answer's name or loosely associated facts is NOT sufficient.
- If the Content omits, contradicts, or fails to address the critical conditions implied by the Question, it should be considered NOT related.

Judgment rules:
1. Identify the key factual constraints in the Question.
2. Check whether the Content provides evidence that supports those constraints in relation to the Answer.
3. If the Answer could reasonably be inferred from the Content under the Question's requirements, return true.
4. Otherwise, return false.

Respond with JSON only in the following format:
{"is_related": true or false, "reason": "concise, evidence-based explanation"}

# Question
<Question>

# Answer
<User agent final wrong answer>

# Content
<Content agent response>

---

## L. Agent Prompts

AgentWebBench employs a two-sided agent architecture with two distinct agent types: user agents and content agents. User agents coordinate information retrieval across multiple websites, while content agents operate autonomously within individual website domains. The prompts for user agents are task-specific, reflecting the different requirements and action spaces across web search, web recommendation, question answering, and deep research tasks. Content agents use a unified prompt design, as they all perform the same core function of retrieving and summarizing documents from their assigned websites.

All prompts follow a structured action-based format, where agents must choose from a predefined set of valid actions (e.g., <search>, <answer>, <summary>) and adhere to strict syntax requirements. This design ensures reproducible agent behavior and enables systematic evaluation of coordination strategies. The prompts emphasize avoiding duplicate searches,

maintaining proper action formatting, and appropriately utilizing retrieved information.

## L.1. User Agent

User agent prompts are designed to guide agents through multi-step information-seeking processes. We mainly follow the prompt design of Chandrahasan et al. (2025) as a foundation, adapting it to AgentWebBench's specific requirements. Each task has distinct objectives and output formats, which are reflected in the corresponding prompts. The prompts include: (a) task-specific role definitions, (b) available websites and their descriptions, (c) valid action specifications with exact syntax requirements, (d) rules for avoiding duplicate operations, and (e) format guidelines for final outputs.

**Web search.** The web search prompt guides the user agent to retrieve and rank relevant documents for a given query. The agent must identify appropriate websites, issue search queries, and return a sorted list of document IDs ranked by relevance. The prompt emphasizes extracting document IDs from search results and formatting the final output as a JSON array.

---

You are a document retrieval assistant with the ability to perform web searches to find relevant documents for a given query. Your task is to retrieve and rank documents, then return a sorted list of document IDs.

You have following websites that can search:
<Website names and descriptions>

Based on the history information, you need to suggest the next action to complete the task.
You will be provided with:
1. Your history search attempts: query in format <search>{"query": query that send to each website, "websites": ["website name", ]} </search>and the returned search results in <information>and </information>. Search results contain documents with their IDs.
2. The query to retrieve documents for.

IMPORTANT: You must strictly adhere to the following rules:
1. Choose ONLY ONE action from the list below for each response, DO NOT perform more than one action per step.
2. Follow the exact syntax format for the selected action, DO NOT create or use any actions other than those listed.
3. **Don't do duplicate search.** Pay attention to the history search results.
4. When you output the final answer, you MUST return a sorted list of document IDs in JSON format. The list should be sorted by relevance (most relevant first).

Valid actions:
1. <search>{"query": query that send to each website, "websites": ["website name", ]} </search>: search the web for documents if you consider you need more information. The search will return documents with their IDs.
2. <answer>["doc_id1", "doc_id2", ...] </answer>: output the final sorted list of document IDs in JSON array format. The document IDs should be sorted by relevance (most relevant first). Extract document IDs from the search results in <information></information>tags.
3. <summary>important parts of the history turns </summary>: summarize the history turns. Reflect the search queries and search results in your history turns, and keep the information you consider important for retrieving relevant documents. Still keep the tag structure, keep search queries between <search>and </search>, and keep search results between <information>and </information>. The history turn information for your subsequent turns will be updated according to this summary action. Don't forget to record relevant document ids in the documents list.

Format:
You should pay attention to the format of your output. You can choose **ONLY ONE** of the following actions:
- If You want to search, You should put the query and candidate websites between <search>and </search>following json format {"query": query that send to each website, "websites": ["website name", ]}.
- If You want to summarize the history turns, You should put the summary between <summary>and </summary>.
- If You want to give the final answer, You should put the sorted list of document IDs in JSON array format between <answer>and </answer>, e.g., <answer>["doc_id1", "doc_id2", "doc_id3"] </answer>. You can only use ONE action per response.

Note:
- Text between <information></information>is the search results from search engine after you perform a search action, **DO NOT** include any information in <information></information>in your output.
- Document IDs in search results appear as part of document metadata. Extract these IDs and return them in a sorted list.
- The final answer must be a valid JSON array of document ID strings, sorted by relevance.

Query: <Query>

---

**Web recommendation.** The web recommendation prompt extends the web search format but requires the agent to first infer the user's next intent from a browsing history sequence. The agent must generate a concise search query that represents what the user would likely search for next, rather than simply rephrasing the history. This task emphasizes intent inference

and query generation before document retrieval.

---

You are a document retrieval assistant with the ability to perform web searches to find relevant documents for a given query. Your task is to retrieve and rank documents, then return a sorted list of document IDs.

You have following websites that can search:
<Website names and descriptions>

Based on the history information, you need to suggest the next action to complete the task.
You will be provided with:
1. Your history search attempts: query in format <search>{"query": query that send to each website, "websites": ["website name", ]} </search>and the returned search results in <information>and </information>. Search results contain documents with their IDs.
2. The query to retrieve documents for.

IMPORTANT: You must strictly adhere to the following rules:
1. Choose ONLY ONE action from the list below for each response, DO NOT perform more than one action per step.
2. Follow the exact syntax format for the selected action, DO NOT create or use any actions other than those listed.
3. **Don't do duplicate search.** Pay attention to the history search results.
4. When you output the final answer, you MUST return a sorted list of document IDs in JSON format. The list should be sorted by relevance (most relevant first).

Valid actions:
1. <search>{"query": query that send to each website, "websites": ["website name", ]} </search>: search the web for documents if you consider you need more information. The search will return documents with their IDs.
2. <answer>["doc_id1", "doc_id2", ...] </answer>: output the final sorted list of document IDs in JSON array format. The document IDs should be sorted by relevance (most relevant first). Extract document IDs from the search results in <information></information>tags.
3. <summary>important parts of the history turns </summary>: summarize the history turns. Reflect the search queries and search results in your history turns, and keep the information you consider important for retrieving relevant documents. Still keep the tag structure, keep search queries between <search>and </search>, and keep search results between <information>and </information>. The history turn information for your subsequent turns will be updated according to this summary action. Don't forget to record relevant document ids in the documents list.

Format:
You should pay attention to the format of your output. You can choose **ONLY ONE** of the following actions:
- If You want to search, You should put the query and candidate websites between <search>and </search>following json format {"query": query that send to each website, "websites": ["website name", ]}.
- If You want to summarize the history turns, You should put the summary between <summary>and </summary>.
- If You want to give the final answer, You should put the sorted list of document IDs in JSON array format between <answer>and </answer>, e.g., <answer>["doc_id1", "doc_id2", "doc_id3"] </answer>.
You can only use ONE action per response.

Note:
- Text between <information></information>is the search results from search engine after you perform a search action, **DO NOT** include any information in <information></information>in your output.
- Document IDs in search results appear as part of document metadata. Extract these IDs and return them in a sorted list.
- The final answer must be a valid JSON array of document ID strings, sorted by relevance.

Query: A user has visited the following sequence of pages:
<User history>
Your task is to infer the user's likely **next intent** based on this browsing history, generate a **concise search query** that represents what they would search for next, and return target documents.

Important:
- Do **not** copy or rephrase the titles.
- **Infer** what the user is looking for, even if it's not explicitly mentioned.

—

Example 1: Browsing history: "iPhone 15 Pro Max review"; "best phone cameras 2024"; "Samsung Galaxy S24 specs"
Search query: flagship phone camera comparison

—

Example 2: Browsing history: "how to brew coffee at home"; "aeropress vs french press"; "best coffee beans for espresso"
Search query: best espresso beans for home brewing

—

---

**Question answering.** The question answering prompt guides the agent to answer factual questions through multi-turn search

and reasoning. Unlike retrieval tasks, the agent must synthesize information from multiple sources to produce concise, factual answers. The prompt emphasizes iterative information gathering and synthesis, allowing the agent to perform multiple searches before generating the final answer.

---

Your are a research assistant with the ability to perform web searches to answer questions. You can answer a question with many turns of search and reasoning.

You have following websites that can search:
<Website names and descriptions>

Based on the history information, you need to suggest the next action to complete the task.
You will be provided with:
1. Your history search attempts: query in format <search>{"query": query that send to each website, "websites": ["website name", ]} </search>and the returned search results in <information>and </information>.
2. The question to answer.

IMPORTANT: You must strictly adhere to the following rules:
1. Choose ONLY ONE action from the list below for each response, DO NOT perform more than one action per step.
2. Follow the exact syntax format for the selected action, DO NOT create or use any actions other than those listed.
3. **Don't do duplicate search.** Pay attention to the history search results.

Valid actions:
1. <search>{"query": query that send to each website, "websites": ["website name", ]} </search>: search the web for information if you consider you lack some knowledge.
2. <answer>answer </answer>: output the final answer if you consider you are able to answer the question. The answer should be short and concise. No justification is needed.
3. <summary>important parts of the history turns </summary>: summarize the history turns. Reflect the search queries and search results in you history turns, and keep the information you consider important for answering the question and generating your report. Still keep the tag structure, keep search queries between <search>and </search>, and keep search results between <information>and </information>. The history turn information for your subsequent turns will be updated according to this summary action.

Format:
You should pay attention to the format of your output. You can choose **ONLY ONE** of the following actions:
- If You want to search, You should put the query and candidate websites between <search>and </search>following json format {"query": query that send to each website, "websites": ["website name", ]}.
- If You want to summarize the history turns, You should put the summary between <summary>and </summary>.
- If You want to give the final answer, You should put the answer between <answer>and </answer>.
You can only use ONE action per response.

Note: text between <information></information>is the search results from search engine after you perform a search action, **DO NOT** include any information in <information></information>in your output.

Question: <Question>

---

**Deep research.** The deep research prompt is the most complex, requiring the agent to write comprehensive research reports in markdown format. The agent can perform multiple actions including planning, searching, writing/revising report scripts, and generating the final report. A critical requirement is proper citation: the agent must cite sources using [n] notation and include a References section at the end, with each citation corresponding to a website that appears in the search results. This task emphasizes multi-step planning, iterative refinement, and proper attribution of information sources.

---

You are a research assistant with the ability to perform web searches to write a comprehensive scientific research article in markdown format. You will be given a question, and you will need to write a report on the question. You can use search tools to find relevant information.
You don't need to write the report in one turn. You can search and revise your report multiple times. When you consider you need some new information, you can perform a search action. When you want to update, generate, or revise your report scripts, you can perform a scripts action. When you consider you have enough information, you can output the final report.

You have following websites that can search:
<Website names and descriptions>

Based on the history information, you need to suggest the next action to complete the task.
You will be provided with:
1. Your history turns information: it might contains your previous plan, report scripts, search results. For search results, queries are in format <search>{"query": query that send to each website, "websites": ["website name", ]} </search>and the returned search

---

results in <information>and </information>.
2. The question to answer.

IMPORTANT: You must strictly adhere to the following rules:
1. Choose ONLY ONE action from the list below for each response, DO NOT perform more than one action per step.
2. Follow the exact syntax format for the selected action, DO NOT create or use any actions other than those listed.
3. **Don't do duplicate search.** Pay attention to the history search results.
4. **Do not always perform the search action. You must consider the history search results and update your report scripts.**
5. When you output the final report in <answer>...</answer>, if you have used information from web search results, you MUST add citations in the form [n] inside the report text.
6. Each [n] MUST correspond to one website (domain) that appears inside <information>...</information>in the history. You are NOT allowed to invent new websites. The same website must always reuse the same [n].
7. At the END of the report (still inside <answer>), add a "References" section in markdown. List each [n] and the corresponding website, for example:
[1] en.wikipedia.org
[2] www.nature.com
8. If there was no web search, or you did not actually use any information from the web results, you SHOULD NOT add any [n] citations or References section.

Valid actions:
1. <search>{"query": query that send to each website, "websites": ["website name", ] } </search>: search the web for information if you consider you lack some knowledge.
2. <plan>plan </plan>: plan the report in your first turn.
3. <scripts>revised or newly generated report scripts </scripts>: revise former report scripts, or newly generate report scripts.
4. <summary>important parts of the history turns </summary>: summarize the history turns. Reflect the plan, scripts, search queries, and search results in you history turns, and keep the information you consider important for answering the question and generating your report. Still keep the tag structure, keep plan between <plan>and </plan>, keep scripts between <scripts>and </scripts>, keep search queries between <search>and </search>, and keep search results between <information>and </information>. The history turn information for your subsequent turns will be updated according to this summary action.
5. <answer>final report </answer>: output the final report.

Format:
You should pay attention to the format of your output. You can choose one of the following actions:
- If You want to search, You should put the query and candidate websites between <search>and </search>following json format {"query": query that send to each website, "websites": ["website name", ]}.
- If You want to make a plan, You should put the plan between <plan>and </plan>.
- If You want to write scripts, You should put the scripts between <scripts>and </scripts>.
- If You want to summarize the history turns, You should put the summary between <summary>and </summary>.
- If You want to give the final report, You should put the report between <answer>and </answer>.
You can only use ONE action per response.

Note: text between <information></information>is the search results from search engine after you perform a search action, **DO NOT** include any information in <information></information>in your output.

Question: <Question>

## L.2. Content Agent

Content agents operate within a single website domain and use a unified prompt design across all tasks. Each content agent is assigned to a specific website and maintains its own retrieval index. The prompt guides the agent to: (a) search within its assigned website using the provided query, (b) retrieve and summarize relevant documents, and (c) return both a natural language summary and a list of up to 3 most relevant document IDs. The unified design ensures consistent behavior across all content agents while allowing them to operate autonomously within their respective domains.

Your are a content assistant with the ability to perform web searches on a specific website "<Website name>" to answer questions. You can answer a question with many turns of search and reasoning.

Website: <Website name>
Website description: <Website description>

Based on the history information, you need to suggest the next action to complete the task.
You will be provided with:
1. Your history search attempts: query in format <search>query </search>and the returned search results in <information>and </information>.

2. The question to answer.

IMPORTANT: You must strictly adhere to the following rules:
1. Choose ONLY ONE action from the list below for each response, DO NOT perform more than one action per step.
2. Follow the exact syntax format for the selected action, DO NOT create or use any actions other than those listed.
3. **Don't do duplicate search.** Pay attention to the history search results.

Valid actions:
1. <search>query </search>: search the web for information if you consider you lack some knowledge.
2. <answer>{"summary": "summary text", "documents": ["doc_id1", "doc_id2", ...]} </answer>: output the final answer if you consider you are able to answer the question. The answer summary should be short and concise. No justification is needed. The documents should be no more than 3 most relevant documents you retrieved from the website and sorted by the document index (most relevant first). If not relevant documents are found, you should return an empty list.
3. <summary>important parts of the history turns </summary>: summarize the history turns. Reflect the search queries and search results in you history turns, and keep the information you consider important for answering the question and generating your report. Still keep the tag structure, keep search queries between <search>and </search>, and keep search results between <information>and </information>. The history turn information for your subsequent turns will be updated according to this summary action. Don't forget to record relevant document ids in the documents list.

Format:
You should pay attention to the format of your output. You can choose **ONLY ONE** of the following actions:
- If You want to search, You should put the query between <search>and </search>.
- If You want to summarize the history turns, You should put the summary between <summary>and </summary>.
- If You want to give the final answer, You should put the answer between <answer>and </answer>.
You can only use ONE action per response.

Note: Text between <information></information>is the search results from search engine after you perform a search action, **DO NOT** include any information in <information></information>in your output.

Question: <Question>

## M. AgentWebBench Websites and Corpus

AgentWebBench is built on the English subset of ClueWeb22-B (Overwijk et al., 2022), a large-scale web corpus collected in 2022 that approximates the most frequently visited portion of the web. From this corpus, we select the 100 websites with the largest document counts, yielding a distributed set of closed-domain sources. These websites collectively contain 18,427,770 documents, with each website assigned to one autonomous content agent. The selected websites span diverse domains including e-commerce, social media, educational platforms, academic repositories, news outlets, and specialized knowledge bases, reflecting the heterogeneous nature of real-world web information sources. Table 12 provides a comprehensive listing of all 100 websites in AgentWebBench, including their names, descriptions, and document counts. The document counts reflect the scale of each website's contribution to the overall corpus, with the largest sources (e.g., Amazon, Wikipedia, Stack Overflow) containing over one million documents each.

**Website descriptions.** Website descriptions are generated using Gemini-2.5-Flash (Comanici et al., 2025) and subsequently checked by human annotators to ensure accuracy and relevance. These descriptions are consumed by user agents in coordination strategies that rely on semantic understanding (e.g., $Tool_P$) to decide which websites should be queried for a given information need.

**Website embeddings.** $Tool_E$ operates on a dense embedding of each website that summarizes its overall topical focus. Instead of using the website description, which is often too general and not sufficiently aligned with specific queries, we construct this embedding from actual site content. For every website, we randomly sample 100 in-domain documents, encode each document into a 1024-dimensional embedding using the MiniCPM-Embedding-Light model (Coelho et al., 2025; MiniCPM Team et al., 2025), and compute the mean of these embeddings to obtain the website representation, which is stored offline. These randomly sampled documents form a small subset of the closed-domain website that does not violate its access constraints, while still providing a representative summary of the website's overall topical focus. At inference time, a user query is embedded into the same vector space and compared with all website embeddings via cosine-similarity-based search, enabling $Tool_E$ to efficiently select semantically aligned websites (top-$k$) for downstream document retrieval. $k$ is set to 3 in our experiments.

**Web-document retrieval tools.** Both $Tool_E$ and $Tool_P$ rely on a non-LLM web-document retrieval tool to fetch documents from the selected websites. This tool is not based on simple regex or keyword matching, but on dense retrieval with semantic

embeddings. For each website, we build an independent dense retrieval index using the same MiniCPM-Embedding-Light model, a state-of-the-art dense retriever trained on 260 million query–document pairs. Every document is encoded into a 1024-dimensional embedding. When a query is issued for a specific website, the tool (a) encodes the query into an embedding, (b) calculates the cosine similarity between the query embedding and the document embeddings, and (c) returns the top-$n$ documents ranked by cosine similarity. This dense retrieval design goes beyond keyword overlap, enabling retrieval of documents that are conceptually related to the query even when they use different surface forms. Maintaining a separate index per website ensures that retrieval is strictly scoped to the corresponding domain, mirroring realistic access constraints in the Agentic Web setting. $n$ is set to 3 in our experiments.

*Table 12.* The 100 websites in AgentWebBench with descriptions and document counts. Websites are selected from ClueWeb22-B based on document count, with each website assigned to one content agent. Descriptions are used by user agents for website selection in coordination strategies.

| Index | Name | Description | #Doc |
|---|---|---|---|
| 1 | www.amazon.com | Welcome to Amazon.com, the world's largest online retailer and a pioneering force in e-commerce. Explore an unparalleled selection of products, from books and electronics to groceries and fashion, delivered conveniently to your door. Beyond shopping, Amazon also offers digital services like streaming media, cloud computing, and smart home devices. | 1,749,804 |
| 2 | en.wikipedia.org | en.wikipedia.org is the English-language version of Wikipedia, a free online encyclopedia that offers a vast and continually growing collection of knowledge. It is a massive collaborative project, with articles contributed and maintained by a global community of volunteers, covering nearly every topic imaginable. | 1,406,613 |
| 3 | stackoverflow.com | Stack Overflow is the definitive online community for developers to learn, share their knowledge, and build their careers. It's a question-and-answer platform where millions of programmers find solutions to coding problems, ask new questions, and contribute their expertise. This vast, collaborative resource is indispensable for anyone working with code. | 1,268,548 |
| 4 | www.youtube.com | YouTube is the world's largest online video-sharing platform, revolutionizing how we consume and create visual content. Millions of users visit daily to watch, upload, and share an incredibly diverse range of videos, from educational tutorials and music videos to vlogs and live streams. It serves as a global hub for creators to connect with audiences and for viewers to discover endless entertainment and information. | 1,181,242 |
| 5 | www.reddit.com | Reddit is a massive network of online communities where users from around the world converge to share content, discuss a myriad of topics, and engage in conversations. Often called "the front page of the internet," it's a dynamic platform where content is upvoted or downvoted by its users, shaping what becomes visible and popular across thousands of specialized subreddits. | 654,802 |
| 6 | www.quora.com | Quora is a leading question-and-answer platform where users can ask questions on virtually any topic and receive answers from a global community. It serves as a vast knowledge-sharing network, allowing individuals to learn, share insights, and connect with others based on shared interests and expertise. | 651,607 |
| 7 | www.imdb.com | IMDb.com is the world's most comprehensive and authoritative source for movie, TV show, and celebrity content. It provides a vast database of information on millions of titles and people, allowing users to explore filmographies, read reviews, watch trailers, and contribute their own ratings. It's an essential resource for filmmakers, fans, and anyone interested in the entertainment industry. | 511,796 |
| 8 | github.com | GitHub is the world's leading platform for software development and version control, leveraging Git to track changes in any set of files. It enables developers to host, review, manage, and collaborate on projects, making it an indispensable tool for open-source contributions and private team collaboration alike. | 481,058 |
| 9 | www.researchgate.net | ResearchGate is the professional network for scientists and researchers worldwide, providing a platform to share papers, ask questions, and discover new research. It connects you with colleagues and enables you to find job opportunities, collaborate on projects, and advance your scientific career. | 412,263 |

| Index | Name | Description | #Doc |
|---|---|---|---|
| 10 | quizlet.com | Quizlet is a popular online learning platform designed to help students and educators master content through various study tools. It primarily offers digital flashcards, practice tests, and interactive games, making learning engaging and effective for any subject. Users can create their own study sets or choose from millions of pre-existing ones. | 392,552 |
| 11 | www.yelp.com | Yelp is your essential online guide for discovering local businesses and reading honest reviews from a vibrant community of users. From restaurants and salons to plumbers and doctors, Yelp helps you make informed decisions by providing detailed business information and customer experiences. It's the perfect tool for exploring what's great in your neighborhood and beyond. | 377,656 |
| 12 | brainly.com | Welcome to Brainly.com, the world's largest online learning platform for students! Here, millions of users connect to ask homework questions and provide expert answers across a vast array of subjects and grade levels. It's a vibrant community dedicated to collaborative learning and academic support. | 372,442 |
| 13 | www.zillow.com | Zillow.com is your ultimate online destination for real estate, offering a comprehensive platform to explore homes for sale, for rent, and estimate property values across the United States. Easily browse millions of listings with detailed information, photos, and virtual tours, making it an essential tool for buyers, sellers, renters, and anyone interested in the housing market. | 366,233 |
| 14 | www.instagram.com | Welcome to Instagram, a leading global social media platform dedicated to visual sharing. Here, you can connect with friends, discover new content, and express yourself by sharing photos and videos. Dive into a vibrant community where stories unfold through images and short clips. | 310,102 |
| 15 | www.sciencedirect.com | ScienceDirect is a premier online platform for scientific, technical, and medical (STM) research, providing access to an extensive collection of peer-reviewed journal articles, books, and other publications. It serves as an essential resource for researchers, students, and professionals worldwide seeking high-quality, authoritative content to support their studies and accelerate discoveries. Explore millions of full-text articles and book chapters across a vast array of disciplines, all from reputable publishers like Elsevier. | 303,906 |
| 16 | pubmed.ncbi.nlm.nih.gov | PubMed, a service of the National Library of Medicine (NLM) at the National Institutes of Health (NIH), is a premier free resource for biomedical literature. It primarily comprises the MEDLINE database, offering millions of citations and abstracts from biomedical and life science journals, along with links to full-text articles. It serves as an essential tool for researchers, clinicians, students, and the public seeking reliable, evidence-based health information. | 251,140 |
| 17 | twitter.com | Twitter.com is a global social media platform where users share short messages called "tweets" to express ideas, provide updates, and engage in public conversations. It's a powerful tool for real-time news, trending topics, and connecting with a diverse range of individuals and organizations worldwide. | 243,146 |
| 18 | www.etsy.com | Etsy is a global online marketplace where creators and collectors connect to buy and sell unique, handmade, and vintage items. It's a vibrant hub for discovering one-of-a-kind treasures, from personalized gifts and artisan crafts to distinctive home decor and clothing. Supporting small businesses and independent artists, Etsy offers a curated alternative to mass-produced goods. | 239,958 |
| 19 | steamcommunity.com | Welcome to Steam Community, the official hub for millions of PC gamers worldwide connected through Valve's Steam platform. Here, you can dive into forums, explore user-created guides, artwork, and screenshots, and engage directly with fellow players and game developers. It's the ultimate place to discuss your favorite titles, discover new content, and enhance your entire gaming experience. | 237,993 |
| 20 | www.linkedin.com | LinkedIn is the world's largest professional networking platform, designed to connect professionals, share knowledge, and foster career growth. Users can create a detailed profile showcasing their experience and skills, discover job opportunities, and build valuable connections within their industry. It serves as a vital tool for career development and professional branding. | 197,809 |

| Index | Name | Description | #Doc |
|---|---|---|---|
| 21 | www.coursehero.com | Course Hero is an online learning platform designed to help students master their courses and excel academically. It provides a vast library of study resources, including course-specific notes, study guides, practice problems, and access to expert tutors. Students leverage Course Hero to deepen their understanding, prepare for exams, and get support with challenging assignments. | 186,126 |
| 22 | www.realtor.com | Realtor.com is a premier online destination for all things real estate, connecting millions with homes for sale, expert agents, and valuable market insights. Whether you're looking to buy your first home, sell your current property, or find a rental, this comprehensive platform offers a wealth of listings and resources. It empowers your entire home journey with tools, data, and direct access to real estate professionals. | 181,980 |
| 23 | www.amazon.co.uk | Welcome to Amazon.co.uk, the United Kingdom's dedicated online marketplace from the global retail giant. Here, you can discover and purchase an immense variety of products, from electronics and books to fashion and groceries, all delivered conveniently to your door. Enjoy competitive prices, extensive product reviews, and reliable shipping services. | 177,647 |
| 24 | www.tripadvisor.com | TripAdvisor is your ultimate global travel guide, helping millions plan and enjoy the perfect trip. Discover hotels, restaurants, attractions, and more through candid reviews, photos, and opinions from a community of fellow travelers. Beyond inspiration, you can also compare prices and book your next adventure directly through the platform. | 168,816 |
| 25 | www.goodreads.com | Goodreads is the world's largest social cataloging website where readers can connect, discover new books, and track their reading journey. You can rate and review titles, create personalized reading lists, and engage with a vibrant community of fellow book lovers. | 167,220 |
| 26 | answers.microsoft.com | Answers.microsoft.com is the official Microsoft support forum, providing a community-driven platform where users can find solutions and get help with a wide range of Microsoft products and services. Here, you can ask questions, browse existing answers, and engage with both experts and fellow users to troubleshoot issues with Windows, Office, Xbox, and more. | 155,961 |
| 27 | www.indeed.com | Indeed.com is one of the world's leading job sites, connecting millions of job seekers with new career opportunities every day. It offers a comprehensive platform where users can search for jobs, post resumes, research companies, and find salary insights. For employers, Indeed provides powerful tools to find and hire the right talent efficiently. | 152,502 |
| 28 | www.slideshare.net | SlideShare is a premier online platform for sharing and discovering presentations, documents, infographics, and videos. It enables professionals, educators, and students to upload and explore content on a wide range of topics, fostering knowledge exchange and professional development. Users can access a vast library of expert-created materials and share their own work with a global audience. | 150,400 |
| 29 | www.yellowpages.com | Discover local businesses and services with ease on Yellow Pages. As a comprehensive online directory, we connect you with everything from restaurants and retailers to plumbers and doctors in your area. Find contact information, directions, reviews, and more to make informed decisions for your everyday needs. | 148,176 |
| 30 | www.ebay.com | eBay is a pioneering global online marketplace that connects millions of buyers and sellers across the world. It offers a vast array of new and used items, ranging from collectibles and electronics to fashion and home goods, available through both auction-style bidding and fixed-price listings. Discover unique finds or easily sell your own items on this dynamic and user-friendly platform. | 146,776 |
| 31 | www.msn.com | MSN.com is Microsoft's long-standing online portal, serving as a comprehensive hub for news, information, and various online services. Users can find a wide array of content, including breaking headlines, financial updates, weather forecasts, and lifestyle articles, all designed to keep them informed and connected throughout their day. | 146,424 |
| 32 | docs.microsoft.com | docs.microsoft.com is the official and comprehensive resource for Microsoft product and technology documentation, offering extensive guides, API references, tutorials, and code examples. It serves as an essential hub for developers, IT professionals, and users seeking in-depth information, support, and learning materials across Microsoft's vast ecosystem. | 140,629 |

| Index | Name | Description | #Doc |
|---|---|---|---|
| 33 | www.homedepot.com | Welcome to homedepot.com, the official online destination for all your home improvement and renovation needs. Explore a vast selection of tools, appliances, building materials, garden supplies, and home decor, along with professional services for any project. It's your go-to resource for transforming your living space, inside and out. | 131,515 |
| 34 | archive.org | Archive.org is a vast digital library dedicated to preserving and providing universal access to human knowledge and cultural artifacts. It offers a massive collection of digitized materials, including the famous Wayback Machine which archives websites, as well as millions of books, audio recordings, videos, software, and much more. This non-profit initiative allows anyone to explore our collective digital heritage. | 129,443 |
| 35 | www.chegg.com | Chegg is a comprehensive online learning platform designed to support students through every stage of their academic journey. It offers a wide range of resources, including textbook rentals, expert-verified homework help, personalized tutoring, and study tools like flashcards and practice problems. With the goal of making education more accessible and effective, Chegg aims to simplify complex subjects and help students achieve their academic goals. | 112,816 |
| 36 | genius.com | Genius.com is the world's largest collection of song lyrics and crowdsourced musical knowledge, offering deep annotations and interpretations for tracks across all genres. Beyond lyrics, it's a vibrant platform where fans and artists alike can dissect music, discuss cultural moments, and explore the stories behind popular songs and other media. | 109,941 |
| 37 | www.redfin.com | Redfin.com is a pioneering real estate website and brokerage dedicated to making buying and selling homes more transparent and efficient. It offers extensive property listings, advanced search tools, and data-driven insights, empowering users with market information and expert agent services. Explore everything from open houses to price history, all designed to streamline your real estate experience. | 106,986 |
| 38 | www.glassdoor.com | Glassdoor is a leading platform that provides unparalleled transparency into the workplace, offering millions of anonymous company reviews, salary reports, and interview insights shared by employees. It empowers job seekers with critical information to make informed career decisions and helps companies build their employer brand. Whether you're researching potential employers or looking for a new role, Glassdoor offers a unique perspective on company culture and compensation. | 104,869 |
| 39 | www.toppr.com | Toppr.com is a comprehensive online learning platform designed to help K-12 students and competitive exam aspirants achieve academic excellence. It provides a vast library of study materials, video lectures, practice questions, and live classes to make learning engaging and effective. Empowering millions, Toppr aims to personalize education and simplify complex concepts for better understanding and top performance. | 103,015 |
| 40 | www.realestate.com.au | Welcome to realestate.com.au, Australia's leading online destination for all your property needs. Explore a vast array of homes for sale, rentals, new developments, and commercial properties across the nation. Whether you're buying, selling, or renting, we provide comprehensive listings, market insights, and expert tools to empower your property journey. | 100,933 |
| 41 | tabs.ultimate-guitar.com | Ultimate Guitar (tabs.ultimate-guitar.com) is the world's largest online resource for guitar tabs, chords, and bass tabs, empowering musicians of all skill levels. Explore a massive, user-contributed library to learn and play your favorite songs with ease, making it an essential tool for guitarists everywhere. | 99,393 |
| 42 | www.merriam-webster.com | Merriam-Webster.com is the definitive online destination for the English language, providing millions of authoritative definitions, pronunciations, and synonyms. As a trusted resource, it offers comprehensive tools for understanding words, exploring vocabulary, and improving writing. | 95,661 |
| 43 | medium.com | Medium.com is a popular online publishing platform where writers and readers connect through compelling stories and insightful articles. It provides a space for diverse voices to share ideas, perspectives, and expertise on a vast array of topics, fostering a community built around engaging content. | 93,035 |

| Index | Name | Description | #Doc |
|---|---|---|---|
| 44 | www.legacy.com | Legacy.com is the leading online destination for obituaries, death notices, and memorials, helping millions connect and remember loved ones. It serves as a vast archive for finding recent and historical tributes, allowing users to share condolences, send flowers, and create lasting legacies. Through its platform, families and friends can celebrate lives, preserve memories, and find comfort in shared remembrance. | 91,972 |
| 45 | www.nature.com | Nature.com is the online hub for Nature Portfolio, a world-leading publisher of cutting-edge scientific research and news. It provides access to the prestigious *Nature* journal and hundreds of other high-impact, peer-reviewed publications across a vast spectrum of scientific disciplines. Explore groundbreaking discoveries, in-depth analysis, and critical insights shaping the future of science. | 90,018 |
| 46 | link.springer.com | Link.springer.com is the comprehensive online platform for Springer Nature's vast collection of academic content. It provides researchers, students, and professionals with access to millions of scientific documents, including journal articles, books, protocols, and reference works across various disciplines, from science and technology to medicine and the humanities. | 88,153 |
| 47 | brainly.in | Brainly.in is a vibrant online learning platform that empowers students to overcome academic challenges. It functions as a collaborative Q&A community where users can ask homework questions across diverse subjects and receive expert answers from peers and educators. This peer-to-peer support system helps students understand concepts and improve their grades. | 87,784 |
| 48 | prezi.com | Prezi is a dynamic presentation software that allows users to create engaging, non-linear presentations on a virtual canvas. Its unique zoomable interface enables fluid storytelling and exploration of ideas, moving beyond traditional slide-by-slide formats. This innovative approach helps captivate audiences and visualize connections between concepts more effectively. | 86,772 |
| 49 | gamefaqs.gamespot.com | Welcome to gamefaqs.gamespot.com, your comprehensive hub for everything gaming. Here, you'll find an extensive library of user-submitted FAQs, walkthroughs, and cheats, complemented by expert game reviews, news, and videos from GameSpot. It's the ultimate resource for players seeking to conquer any challenge. | 86,398 |
| 50 | www.theguardian.com | Welcome to 'www.theguardian.com', your source for award-winning independent journalism. Discover in-depth global news and analysis spanning politics, culture, business, and current affairs. It's a trusted platform committed to challenging powerful institutions and informing readers worldwide. | 84,935 |
| 51 | www.ebay.co.uk | eBay.co.uk is the UK's premier online marketplace, connecting millions of buyers and sellers for a vast array of goods. From brand new electronics and fashion to unique collectibles and pre-owned treasures, you can find almost anything you're looking for. It's the go-to platform for discovering great deals and conveniently selling items across the United Kingdom. | 84,736 |
| 52 | www.nexusmods.com | Welcome to Nexus Mods, the world's largest site for PC game modifications. Discover and download an incredible variety of user-created content to enhance, customize, and extend the lifespan of thousands of your favorite titles. Join a passionate community dedicated to exploring and expanding the possibilities within gaming. | 84,532 |
| 53 | www.bbc.co.uk | BBC.co.uk is the official website for the British Broadcasting Corporation, a world-renowned public service broadcaster. It serves as a comprehensive portal, offering breaking news, in-depth analysis, live radio and TV streams, and a vast archive of entertainment, educational, and cultural content. Explore a trusted source for UK and global stories, sports, documentaries, dramas, and much more. | 81,449 |
| 54 | www.pinterest.com | Pinterest is a visual discovery engine designed to help you find inspiration for all aspects of your life. Users can discover, save, and organize ideas—known as "Pins"—onto personalized boards across a vast array of interests, from recipes and home decor to fashion and travel. It's the perfect place to explore new hobbies, plan projects, and curate your future aspirations. | 80,682 |
| 55 | www.answers.com | Answers.com is a comprehensive online resource dedicated to providing quick and reliable information across a vast array of topics. It serves as a user-friendly platform where questions are answered by a community of contributors, offering insights from general knowledge to specific how-to guides. Explore its extensive database to find the information you need, whether for learning, problem-solving, or general curiosity. | 79,675 |

| Index | Name | Description | #Doc |
|---|---|---|---|
| 56 | www.dnb.com | Welcome to DNB.com, the official online platform for Dun & Bradstreet. Here, businesses access critical data, analytics, and insights essential for managing risk, identifying opportunities, and driving growth. Leverage D&B's vast information to make informed decisions and build stronger relationships worldwide. | 77,950 |
| 57 | www.wordplays.com | Welcome to Wordplays.com, your ultimate online resource for all things word-related! Whether you're looking for Scrabble help, crossword answers, or just want to explore anagrams and definitions, our comprehensive tools are designed to enhance your word game experience. Dive into our extensive database to find the perfect word for any occasion. | 76,671 |
| 58 | www.rightmove.co.uk | Welcome to Rightmove, the UK's leading property website. Here, you can explore an extensive database of homes for sale and to rent across the entire United Kingdom. Whether you're searching for your dream house or the perfect rental, Rightmove connects you with thousands of listings and local agents. | 74,841 |
| 59 | www.mapquest.com | MapQuest.com is a pioneering online mapping service, offering users detailed maps and reliable, turn-by-turn directions. It empowers you to easily find locations, plan routes for driving or walking, and navigate effectively to any destination. This classic platform continues to be a go-to resource for trip planning and getting where you need to go. | 74,600 |
| 60 | www.target.com | Welcome to Target.com, the official online destination for one of America's most beloved retail chains. Explore a vast selection of products, from everyday essentials and groceries to fashion, home decor, electronics, and beyond. Enjoy convenient online shopping, discover the latest deals, and manage your Target experience all in one place. | 73,410 |
| 61 | www.gov.uk | Welcome to GOV.UK, the official website for the UK government. This central platform provides comprehensive public services and information from various government departments, agencies, and public bodies. It's your authoritative source for everything from paying taxes to applying for a passport. | 73,268 |
| 62 | www.accuweather.com | AccuWeather.com is a leading global source for highly accurate and localized weather forecasts. It provides real-time updates, detailed hourly and daily predictions, radar maps, and severe weather alerts for millions worldwide. Whether you're planning your day or monitoring conditions for business, AccuWeather offers comprehensive and reliable weather information. | 72,188 |
| 63 | www.azlyrics.com | AZLyrics.com is a premier online resource for an extensive database of song lyrics from countless artists and genres. It allows users to easily find the words to their favorite songs, meticulously organized alphabetically by artist and then by song title. Whether you're looking to sing along, understand the meaning behind a track, or simply explore lyrical content, AZLyrics provides quick and reliable access. | 71,294 |
| 64 | www.trulia.com | Trulia is a premier online real estate platform dedicated to helping you find your next home, whether you're buying, selling, or renting. Explore extensive property listings, detailed neighborhood information, and local market trends to make informed decisions. It provides tools and insights to simplify your real estate journey. | 69,188 |
| 65 | www.academia.edu | Academia.edu is a leading social networking platform for academics and researchers worldwide, allowing them to share scholarly papers, monitor their impact, and connect with peers. It serves as a valuable resource for disseminating preprints, articles, and presentations, fostering collaboration and the discovery of new research across disciplines. | 68,955 |
| 66 | www.scribd.com | Scribd is a comprehensive digital library and subscription service offering unlimited access to millions of ebooks, audiobooks, magazines, documents, and sheet music. It provides a vast platform for readers and learners to discover and engage with a diverse range of content across various genres and subjects. | 67,730 |
| 67 | www.thefreedictionary.com | TheFreeDictionary.com is a comprehensive online resource providing instant access to definitions, a thesaurus, encyclopedias, and a wide array of specialized dictionaries. It's an invaluable tool for students, writers, and anyone looking to quickly understand words, explore synonyms, or research topics across various fields. Its user-friendly interface makes it a go-to for quick and reliable information. | 66,984 |

| Index | Name | Description | #Doc |
|-------|------|-------------|------|
| 68 | study.com | Study.com is a comprehensive online learning platform dedicated to making education accessible and engaging. It offers a vast library of video lessons, courses, and test prep materials across numerous subjects and academic levels, from K-12 to college and professional development. Whether you're seeking homework help, college credit, or career advancement, Study.com provides flexible resources to support your learning journey. | 65,705 |
| 69 | health.usnews.com | Welcome to health.usnews.com, the definitive health section of U.S. News & World Report, offering comprehensive and reliable information on a wide array of health topics. Explore their renowned hospital and diet rankings, delve into expert advice, stay updated with the latest health news, and utilize practical tools designed to empower your well-being decisions. | 64,666 |
| 70 | www.healthgrades.com | Healthgrades.com is a leading online resource dedicated to helping you find and connect with healthcare providers. It offers comprehensive profiles of doctors, dentists, and hospitals, featuring patient reviews, ratings, and vital practice information. This platform empowers you to make informed decisions about your medical care by providing transparency and accessibility to provider data. | 63,949 |
| 71 | apps.apple.com | Apps.apple.com is the official online gateway to the Apple App Store, your ultimate destination for millions of applications. Explore a vast collection of apps for iPhone, iPad, Mac, Apple Watch, and Apple TV, covering everything from productivity to entertainment. Easily discover new tools, games, and services, read reviews, and download directly to enhance your Apple device experience. | 62,951 |
| 72 | www.forbes.com | Forbes.com is the digital platform for the world-renowned Forbes magazine, a leading source for news and insights on business, finance, and entrepreneurship. It offers a comprehensive look into the worlds of wealth, technology, leadership, and innovation, serving as an essential resource for executives, investors, and aspiring entrepreneurs globally. | 61,549 |
| 73 | answers.sap.com | Welcome to answers.sap.com, your go-to community for all things SAP! Here, you can find solutions to your questions, share knowledge, and connect with a vast network of SAP experts and users worldwide. Dive in to explore discussions, troubleshoot issues, and enhance your understanding of SAP products and technologies. | 60,839 |
| 74 | www.discogs.com | Discogs.com is the world's largest music database, meticulously cataloging millions of releases across all formats and genres. It also serves as a vibrant marketplace where users can buy and sell physical music, from vinyl and CDs to cassettes, fostering a global community of collectors and enthusiasts. | 60,401 |
| 75 | www.wordhippo.com | WordHippo.com is your comprehensive online dictionary and thesaurus, designed to empower your vocabulary and writing. Discover a vast array of word tools including synonyms, antonyms, definitions, rhymes, translations, word forms, and more, all readily accessible in one user-friendly platform. | 60,211 |
| 76 | www.findagrave.com | Find a Grave is a comprehensive online database dedicated to cemeteries and burial records worldwide. It allows users to search for ancestors' grave locations, view photos of headstones, and contribute information or create virtual memorials. This crowdsourced platform serves as an invaluable resource for genealogy research and preserving the legacy of individuals. | 59,855 |
| 77 | screenrant.com | Screen Rant is a leading online destination for entertainment news, reviews, and insightful analysis across movies, television, and comic books. Dive into their extensive coverage for the latest industry updates, in-depth breakdowns of pop culture phenomena, and engaging fan theories. It's the ultimate resource for enthusiasts looking to stay informed and explore the vast world of cinema, TV, and beyond. | 59,384 |
| 78 | www.manualslib.com | Welcome to ManualsLib.com, your comprehensive online library for product manuals. Easily find, view, and download user guides for a vast array of electronics, appliances, and other devices, ensuring you always have the instructions you need. Simplify troubleshooting and setup with instant access to detailed information right at your fingertips. | 58,611 |
| 79 | www.bbb.org | The Better Business Bureau (BBB) is a non-profit organization dedicated to fostering an ethical marketplace where buyers and sellers can trust each other. It provides free business reviews and ratings, helping consumers make informed purchasing decisions and find trustworthy companies. The BBB also helps businesses build credibility and resolve customer complaints, promoting higher standards of business practice. | 57,094 |

| Index | Name | Description | #Doc |
|---|---|---|---|
| 80 | www.nytimes.com | www.nytimes.com is the official digital home of The New York Times, a globally recognized leader in journalism. It provides comprehensive and in-depth coverage of breaking news, politics, business, culture, and more, alongside award-winning investigative reports and diverse opinion pieces. Readers can access a vast archive and up-to-the-minute analysis, establishing it as a cornerstone for authoritative information worldwide. | 57,068 |
| 81 | doctor.webmd.com | Welcome to doctor.webmd.com, your dedicated resource for connecting with healthcare professionals. Here, you can easily search for doctors by specialty, location, and insurance, access detailed physician profiles, and read patient reviews to help you make informed decisions about your care. | 56,878 |
| 82 | www.amazon.in | Welcome to Amazon India, your go-to online marketplace for an unparalleled shopping experience. Discover an extensive range of products, from everyday essentials and cutting-edge electronics to fashion and home decor, all delivered conveniently to your doorstep. Enjoy competitive prices, reliable delivery, and a user-friendly interface designed to make your online shopping effortless. | 56,701 |
| 83 | www.lowes.com | Lowe's is a leading home improvement retailer, offering a vast selection of products for every aspect of your home. From appliances and tools to building materials, garden supplies, and decor, you'll find everything you need for repairs, renovations, and new projects. Explore their extensive catalog and get expert advice to transform your living spaces. | 56,286 |
| 84 | www.dailymail.co.uk | DailyMail.co.uk is the official online platform for the British newspaper, the Daily Mail. It offers a comprehensive blend of breaking news, politics, and current affairs, alongside extensive coverage of celebrity gossip, lifestyle features, and opinion pieces. As one of the world's most-read English-language news websites, it provides a distinct perspective on both global and UK events. | 55,472 |
| 85 | www.britannica.com | Britannica.com is the online home of the Encyclopædia Britannica, a world-renowned and authoritative source of information. It offers expertly curated articles, images, and videos covering a vast range of subjects, from history and science to arts and culture. As a trusted resource for students, researchers, and curious minds, it provides in-depth, accurate knowledge for everyone. | 53,997 |
| 86 | www.twitch.tv | Twitch.tv is the world's leading live streaming platform, primarily renowned for its massive gaming community but also hosting a wide array of content like creative arts, music, and "Just Chatting." It enables creators to broadcast themselves in real-time, fostering interactive communities where viewers can engage directly with streamers and fellow fans through live chat and support. | 52,147 |
| 87 | www.wikihow.com | WikiHow is a leading online platform that offers an extensive collection of step-by-step guides on virtually any topic imaginable. From practical life skills to creative hobbies, its articles provide clear instructions, often accompanied by illustrations, to help you learn "how to" do almost anything. It's your go-to resource for mastering new skills, solving problems, or simply satisfying your curiosity. | 52,123 |
| 88 | www.food.com | Welcome to Food.com, your ultimate online destination for culinary inspiration and delicious recipes! Explore a vast collection of user-submitted and editor-tested dishes, from quick weeknight meals to impressive holiday feasts. Join a vibrant community of home cooks, share your creations, and discover new favorites that will delight your taste buds. | 51,944 |
| 89 | www.drugs.com | Drugs.com is a leading online resource providing comprehensive, accurate, and up-to-date information on medications. It offers detailed drug profiles, covering uses, dosages, side effects, and drug interactions, along with a wealth of information on various health conditions. This invaluable tool helps patients, caregivers, and healthcare professionals make informed decisions about prescription drugs, over-the-counter medicines, and supplements. | 50,787 |
| 90 | www.maxpreps.com | MaxPreps.com is the definitive digital hub for high school sports across the United States. It offers comprehensive coverage, including scores, schedules, statistics, rankings, and news for thousands of teams and athletes in various sports. From football and basketball to soccer and volleyball, MaxPreps connects the high school sports community with up-to-the-minute information and highlights. | 50,223 |

| Index | Name | Description | #Doc |
|---|---|---|---|
| 91 | www.bestbuy.com | Welcome to BestBuy.com, your premier online destination for the latest in electronics, appliances, and innovative technology. Explore a vast selection of top brands, find expert advice, and discover everything you need to enhance your home, work, and entertainment experiences. From cutting-edge gadgets to essential home appliances, BestBuy.com connects you with the best in tech. | 50,214 |
| 92 | www.deviantart.com | DeviantArt is the world's largest online social community for artists and art enthusiasts. It provides a vibrant platform for creators to showcase their original artwork, find inspiration, and connect with a global audience. Users can explore a vast array of artistic expressions, from digital art and photography to traditional mediums and fan art. | 50,186 |
| 93 | www.amazon.ca | Welcome to Amazon.ca, the premier online shopping destination for Canadians! Explore millions of products across every category imaginable, from electronics and books to fashion and groceries, all delivered right to your door. Enjoy competitive prices, convenient shipping options, and a seamless shopping experience tailored for Canadian customers. | 49,967 |
| 94 | tvtropes.org | TV Tropes is a wiki dedicated to cataloging and dissecting the narrative conventions and devices—known as 'tropes'—that appear in fiction across all mediums. It offers deep dives into recurring patterns, character archetypes, and plot mechanisms, helping users understand and identify storytelling techniques in everything from film to literature and video games. | 49,864 |
| 95 | www.geeksforgeeks.org | GeeksforGeeks is a leading online portal dedicated to computer science and programming education. It provides a comprehensive collection of articles, tutorials, interview questions, and coding challenges for students and professionals alike. This platform is an indispensable resource for anyone looking to master technical concepts and excel in their careers. | 49,585 |
| 96 | community.spiceworks.com | Welcome to the Spiceworks Community, the premier online hub for IT professionals worldwide. Here, you can connect with peers, find solutions to technical challenges, share your expertise, and stay updated on the latest in IT. It's an invaluable resource for anyone working in IT to collaborate and grow. | 47,110 |
| 97 | www.allrecipes.com | Welcome to Allrecipes.com, your go-to online destination for a vast collection of user-submitted and professionally tested recipes. Explore millions of dishes, read reviews from fellow home cooks, and find endless inspiration for your next meal or culinary adventure. It's the ultimate community for sharing, discovering, and mastering delicious recipes from around the world. | 47,098 |
| 98 | www.dell.com | Welcome to Dell.com, your premier destination for cutting-edge technology solutions. Explore a comprehensive range of high-performance laptops, desktops, monitors, and accessories, alongside powerful servers and IT infrastructure. Discover innovation crafted to empower your personal, professional, and gaming experiences. | 46,748 |
| 99 | www.ncbi.nlm.nih.gov | The National Center for Biotechnology Information (NCBI) is a leading resource provided by the U.S. National Library of Medicine (NLM), part of the National Institutes of Health (NIH). It serves as a comprehensive hub for biomedical and genomic information, offering access to an extensive range of scientific databases, research tools, and peer-reviewed literature for researchers and the public worldwide. | 46,528 |
| 100 | download.cnet.com | Download.cnet.com is a premier destination for discovering and downloading a vast array of software, apps, and drivers. It offers a trusted source for free and trial versions, often accompanied by CNET's expert reviews and user ratings to help you make informed choices. | 46,510 |

