# OpenReview forum: "AgentWebBench: Benchmarking Multi-Agent Coordination in Agentic Web"
_ICML.cc/2026/Conference — ICML 2026 regular_

### Official Review · Reviewer_73Gu · 2026-03-11

**Soundness:** 3
**Presentation:** 3
**Significance:** 4
**Originality:** 3
**Overall Recommendation:** 4
**Confidence:** 4

**Summary:**

This paper introduces AgentWebBench, a benchmark for studying multi-agent coordination in an “Agentic Web” setting where a user agent must query website-specific content agents rather than directly accessing a centralized corpus. It includes four tasks—web search, web recommendation, QA, and deep research—over 100 websites and 18.4M documents, and compares several coordination strategies. A central finding is that decentralized coordination usually trails centralized retrieval, though the gap narrows with stronger models and can reverse for QA.

**Compliance With Llm Reviewing Policy:**

Affirmed.

**Final Justification:**

The rebuttal clears my questions, but I don't believe that the paper is worthing a spotlight or oral, so I keep my score unchanged.
Overall, the paper is acceptable.

**Key Questions For Authors:**

How representative is the current corpus construction of the broader future Agentic Web?

**Limitations:**

yes

**Strengths And Weaknesses:**

This is an ambitious and timely benchmark paper with a fresh problem formulation. I particularly like the attempt to operationalize the idea of a decentralized web of content agents, since this is a realistic direction for future agent ecosystems. The benchmark is large in scale, covers multiple task types, and goes beyond leaderboard reporting by analyzing coordination strategies, efficiency, and broader systemic properties such as traffic concentration and failure modes.

However, there are still some drawbacks：
1. The web recommendation task is currently too underpowered to support strong conclusions, because the paper itself reports that “most methods have near-zero recall” on this setting. When nearly all systems collapse to the floor, it becomes hard to tell whether the benchmark is measuring coordination quality or simply reflecting a task setup that current methods cannot meaningfully engage with.

2. The paper’s current comparisons do not fully isolate coordination from retrieval design, because the three strategies differ along both website selection and document retrieval. As stated in Section 3.2, the methods change not only whether agents communicate, but also how websites are selected and whether retrieval is tool-based or agent-mediated. Therefore, when Multi-Agent differs from ToolE or ToolP, the gain cannot be attributed cleanly to coordination alone.

3. The paper provides evidence that performance differences are task-dependent, but the causal explanation remains incomplete. For example, it shows that ToolP is often stronger than Multi-Agent on web search, while the gap is less visible on deep research, and it argues that content agents are less stable on retrieval-heavy tasks. However, the analysis stops short of explaining which component causes this difference—e.g., content-agent summarization noise, retrieval instability, or interaction overhead. As a result, the reader learns that task dependence exists, but not yet why it emerges.

---

> ### Author Rebuttal · Authors · 2026-03-30
>
> > ### **Q: Corpus representativeness**
>
> We thank the reviewer for this important question. We provide a detailed discussion of the same concern, including gaps and future extensions, in our response to Reviewer `tMi6` **Q1&W1**.
>
> > ### **W1: Web recommendation is underpowered**
>
> The result is not underpowered. The web recommendation task is adapted from ORBIT (He et al., 2025), an established web recommendation benchmark. It is an existing set up of general web recommendations, and focuses on the long tail cold start part.
>
> **(1) Near-zero scores are consistent with ORBIT's results, reflect intrinsic task difficulty.** On ORBIT, even state-of-the-art LLMs achieve NDCG@10 below 1% on candidate ranking (`Appendix I, Table 10`). This is because user browsing histories span unrelated domains (e.g., medical, gaming, music, legal) yet the target page is hard to predict (`Appendix I, Case`). QueryGen (He et al., 2025), a state-of-the-art non-agent method, also reaches near-zero scores on our corpus (`Appendix I, Table 9`). These results confirm that the difficulty is intrinsic to this task.
>
> **(2) Even with near-zero scores, the task provides insights.** For example, our failure attribution (`Sec. 5.5, Table 5`) shows that user/content-agent failure rates on web search vs. web recommendation shows different patters across seven LLMs, revealing the insight that in agentic web, different tasks need different optimization.
>
> > ### **W2: Cannot cleanly isolate coordination from retrieval**
>
> **(1) Progressive design.** We agree that the three strategies do not follow a strict controlled-variable design. Our design is progressive: Tool_E → Tool_P → Multi-Agent introduces one new capability at each step. Tool_E uses embedding-based website selection with a content tool for retrieval; Tool_P introduces LLM reasoning for website selection while keeping the same content tool; Multi-Agent replaces the content tool with autonomous content agents for agent-to-agent communication. This progression reflects how the Agentic Web is likely to evolve in practice.
>
> However, we acknowledge that this progressive design has limited ability to isolate individual factors. To address this concern, we have conducted below experiments that complete a factorial design.
>
> **(2) Additional experiments.** To fully isolate these factors, we conducted additional experiments that complete a 2×2 factorial design by adding a missing configuration (Tool_E + Content Agent):
>
> | Method, Gemini-3 | Website Selection | Document Retrieval | N@3 (Web Search) | Acc. (QA) |
> |---|---|---|---|---|
> | Tool_E | Embedding | Content Tool | 39.56 | 56.60 |
> | Tool_P | LLM Reasoning | Content Tool | 47.52 | 67.92 |
> | Tool_E + Content Agent | Embedding | Content Agent | 37.88 | 58.49 |
> | Multi-Agent | LLM Reasoning | Content Agent | 44.34 | 67.92 |
>
> Comparing rows with the same Website Selection isolates Document Retrieval effects, while comparing rows with the same Document Retrieval isolates Website Selection effects. The **key findings remain consistent**:
> - LLM reasoning for website selection consistently improves performance (Tool_E → Tool_P and Tool_E+Content Agent → Multi-Agent)
> - Content agent retrieval introduces modest overhead on web search but remains competitive on QA.
>
> We will include these results in the revision.
>
> > ### **W3: Incomplete causal explanation for task-dependent performance**
>
> We synthesize them into a more explicit causal account for existing analyses.
>
> The key difference is that Tool_P retrieves documents via a deterministic tool call, while Multi-Agent delegates retrieval to content agents that reason autonomously. This difference interacts with task characteristics in three ways:
>
> **(1) Autonomous reasoning introduces noise on precise tasks.** Content agents may reformulate short, precise web search queries in ways that drift from the original intent (`Appendix H`). On QA and deep research, the same reasoning helps by iteratively refining retrieval and filtering irrelevant results.
>
> **(2) Multi-turn interaction trades overhead for coverage.** Each additional round carries a risk of failure (unparseable requests or empty returns, as measured by the validity rate in `Sec. 5.3`). For web search, where a single retrieval often suffices, this overhead adds risk without proportional gain. For QA and deep research, additional rounds provide genuine value through adaptive query refinement and broader source coverage.
>
> **(3) Output granularity determines error tolerance.** Web search requires returning specific ranked documents, so retrieval imprecision directly harms scores. QA and deep research require synthesized answers, where partial evidence from multiple rounds can still contribute to correct outputs.
>
> These three factors explain the empirical pattern: the Tool_P vs. Multi-Agent gap is largest on web search, disappears on deep research, and reverses on QA. We will expand this analysis in the revision.

---

> > ### Author Rebuttal · Reviewer_73Gu · 2026-04-03
> >
> > Issue Resolved

---

> > > ### Author Response · Authors · 2026-04-03
> > >
> > > Dear Reviewer 73Gu,
> > >
> > > Thank you for your thoughtful and constructive feedback. We are glad to hear that our rebuttal has fully resolved your concerns.
> > >
> > > We would greatly appreciate it if you could consider adjusting your score accordingly. Should any questions remain, we are happy to discuss further.
> > >
> > > Best,
> > > Authors

---

### Official Review · Reviewer_amui · 2026-03-12

**Soundness:** 2
**Presentation:** 3
**Significance:** 2
**Originality:** 3
**Overall Recommendation:** 3
**Confidence:** 4

**Summary:**

The authors present AgentWebBench, a benchmark for studying information access in a decentralized “Agentic Web” setting where a user agent cannot directly access a global corpus and must instead coordinate with website-specific content agents. The benchmark is built from the English subset of ClueWeb22-B and contains 100 websites with 18.4M documents. It evaluates four tasks—web search, web recommendation, question answering, and deep research—under three coordination strategies: ToolE (embedding-based website selection), ToolP (LLM-based website selection), and Multi-Agent (agent-to-agent communication with autonomous content agents). The paper evaluates these settings across seven LLMs and compares them to centralized baselines. My main concern about soundness is the realism gap between the paper’s broad “Agentic Web” framing and the actual benchmark instantiation. Although the benchmark is useful as a controlled environment, all content agents are built over the same corpus family and the same type of dense retrieval setup, rather than genuinely heterogeneous provider-controlled systems with different APIs, policies, ranking objectives, authentication constraints, or latency profiles. That makes the benchmark a good abstraction, but not yet a strong proxy for the broader ecosystem claims.

**Compliance With Llm Reviewing Policy:**

Affirmed.

**Final Justification:**

The final claim is so narrow that scores had to be adjusted/lowered.

**Key Questions For Authors:**

-

**Limitations:**

yes

**Strengths And Weaknesses:**

From an originality standpoint, this is a timely and fairly novel benchmark direction. The authors investigate the concept of evaluating information access when coordination with provider-controlled agents replaces direct centralized retrieval, which is a meaningful shift from standard web-agent and RAG benchmarks. The benchmark setup is also reasonably broad: four tasks, multiple coordination strategies, and a nontrivial corpus scale.

---

> ### Author Rebuttal · Authors · 2026-03-30
>
> We thank the reviewer for recognizing AgentWebBench as a "timely and fairly novel benchmark direction" with a "meaningful shift from standard web-agent and RAG benchmarks."
>
> > ### **Q: My main concern about soundness is the realism gap between the paper’s broad “Agentic Web” framing and the actual benchmark instantiation. Although the benchmark is useful as a controlled environment, all content agents are built over the same corpus family and the same type of dense retrieval setup, rather than genuinely heterogeneous provider-controlled systems with different APIs, policies, ranking objectives, authentication constraints, or latency profiles. That makes the benchmark a good abstraction, but not yet a strong proxy for the broader ecosystem claims.**
>
> The current setup of our benchmark is a deliberate design choice for two reasons.
>
> **(1) Controlled evaluation as a first step.** A uniform retrieval setup across all content agents is the first step to isolate the effect of coordination strategies from infrastructure variation. This follows standard benchmark methodology: BEIR (Thakur et al., 2021) fixes the retrieval model across 18 datasets to isolate cross-domain generalization, and WebArena (Zhou et al., 2024) standardizes web environments to isolate agent planning.
>
> **(2) Implicit heterogeneity already exists.** Although the retrieval infrastructure is shared, the 100 websites differ substantially in content distribution, topic coverage, document quality, and domain specificity (`Appendix L`). These differences naturally create heterogeneous coordination challenges. For example, querying wikipedia.org vs. sciencedirect.com vs. instagram.com requires different query formulations and result interpretation strategies, which is reflected in the traffic concentration patterns we observe (`Sec. 5.2`).
>
> **(3) Additional ablation study.** To directly address this concern, we have conducted additional experiments using BM25 and hybrid retrieval as alternatives to dense retrieval on the web search task with Gemini-3:
> - BM25: Keyword-based retrieval using BM25, with 82,860,270 unique terms extracted from 18,427,770 documents.
> - Hybrid: Reciprocal Rank Fusion combining dense and BM25 retrieval.
>
> | Retrieval Model | Method | N@3 | N@5 | R@3 | R@5 |
> |---|---|---|---|---|---|
> | Dense (Ours) | Classic IR | 47.86 | 51.04 | 57.63 | 65.25 |
> | Dense (Ours) | Classic | 52.21 | 53.77 | 60.17 | 63.84 |
> | Dense (Ours) | Tool_E | 39.56 | 40.49 | 44.63 | 46.89 |
> | Dense (Ours) | Tool_P | 47.52 | 49.07 | 56.50 | 60.17 |
> | Dense (Ours) | Multi-Agent | 44.34 | 47.08 | 51.98 | 58.47 |
> | BM25 | Classic IR | 28.12 | 31.48 | 33.33 | 41.53 |
> | BM25 | Classic | 38.47 | 38.58 | 42.37 | 42.66 |
> | BM25 | Tool_E | 29.73 | 29.84 | 32.20 | 32.49 |
> | BM25 | Tool_P | 35.75 | 36.93 | 40.40 | 43.22 |
> | BM25 | Multi-Agent | 33.07 | 34.93 | 38.98 | 43.50 |
> | Hybrid | Classic IR | 43.44 | 47.41 | 51.98 | 61.86 |
> | Hybrid | Classic | 50.53 | 51.47 | 58.19 | 60.45 |
> | Hybrid | Tool_E | 37.59 | 37.95 | 42.09 | 42.94 |
> | Hybrid | Tool_P | 45.35 | 47.21 | 52.82 | 57.34 |
> | Hybrid | Multi-Agent | 44.94 | 46.35 | 53.11 | 56.50 |
>
> Across all three retrieval backends, our key findings are consistent:
> - Tool_P is often better than Tool_E (`Section 5.1 (2)`), which suggests that LLM reasoning helps pick more relevant websites than embedding similarity.
> - Content agents are less stable on retrieval-heavy tasks (`Section 5.1 (3)`). Tool_P and Multi-Agent mainly differ in whether they use content agents. Tool_P is usually stronger, which suggests the current content-agent retrieval strategy is not yet stable for retrieval-focused tasks.
>
> This confirms that our conclusions are robust to retrieval backend choice. We will include these results in the revision.
>
> We thank the reviewer for raising this important concern. Extending the benchmark to support heterogeneous agentic web and broader agentic tasks is a promising future direction that our modular decentralized design naturally supports.

---

> > ### Author Rebuttal · Reviewer_amui · 2026-04-04
> >
> > The rebuttal improves the paper, but it does not resolve my main concern. The added BM25/hybrid ablation is useful and shows the trends are not tied to one retrieval backend. However, my original concern was the realism of the benchmark relative to the paper’s broad "Agentic Web" framing. The setup still uses a largely standardized infrastructure across websites and does not capture key provider-side heterogeneity such as differing interfaces, policies, objectives, access constraints, or reliability/latency characteristics. As a result, the benchmark is a useful controlled abstraction, but not yet a strong proxy for the broader ecosystem the paper claims to study. Addressing this requires more than a short rebuttal, either through substantial benchmark changes or materially narrowing the claims. If this is not addressed in the revision, it will lower my final score.

---

> > > ### Author Response · Authors · 2026-04-06
> > >
> > > We thank the reviewer for the constructive feedback and for acknowledging that the retrieval ablation demonstrates robustness across retrieval backends. We take the remaining concern seriously and commit to addressing it through concrete revisions.
> > >
> > >
> > > The reviewer raises an important point: the current benchmark is a controlled abstraction, and our framing should not imply broader ecological validity than the setup delivers. We fully accept this criticism and will narrow the claims to precisely match the scope of our experimental setup, as outlined below.
> > >
> > >
> > > ## **Concrete revisions we commit to in the paper.**
> > >
> > >
> > > **(1) Narrowing the framing.** We will revise the abstract, introduction including novelties, and related work to explicitly scope the benchmark as a controlled testbed for studying coordination under decentralized access constraints, rather than a simulation of the full Agentic Web ecosystem. Below is our revised abstract draft, with changes **bolded** relative to the original:
> > >
> > > > Agentic Web is an emerging paradigm where autonomous agents help users use online information. As the paradigm develops, content providers are also deploying agents to manage their data and serve it through controlled interfaces. This shift moves information access from centralized retrieval to decentralized coordination. **As a first step toward studying this paradigm under controlled conditions,** we introduce AgentWebBench, a benchmark that evaluates how well a user agent synthesizes answers by interacting with website-specific content agents. **All content agents share the same retrieval infrastructure, allowing the benchmark to isolate coordination behavior from provider-side variation.** We evaluate four tasks that cover common web information needs, spanning ranked retrieval (web search, web recommendation) and open-ended synthesis (question answering, deep research). Across seven advanced LLMs and three coordination strategies, multi-agent coordination generally lags behind centralized retrieval as expected, because user agent cannot directly access the corpus, but the gap shrinks with model scale and can even outperform centralized retrieval on question answering. **Within this controlled setup,** we find that decentralized access concentrates traffic toward a small set of websites, test time scaling improves both interaction reliability and task performance, and strong results require sufficient interactions guided by careful planning. Finally, our failure analysis suggests that user agents need better planning and answer synthesis, while content agents need more reliable retrieval and evidence quality. **Extending the benchmark to heterogeneous provider-controlled systems with diverse interfaces, policies, and ranking objectives is an important direction for future work.** Code, data, and APIs will be released online.
> > >
> > >
> > > **(2) Adding a "Scope and Limitations" section to the main body.** We will add a dedicated Limitations section in the main paper. This section will clearly list what the benchmark does not model. Below is the draft:
> > >
> > > > **Scope and Limitations.** AgentWebBench is designed as a controlled testbed under decentralized access constraints. To achieve this, we deliberately fix the retrieval infrastructure and unified instruction template across all content agents: every website uses the same dense retrieval backend and returns results in a uniform format. This design parallels established benchmark methodology and enables fair comparison across coordination strategies without website design bias. However, this controlled setup does not capture several dimensions of real-world provider heterogeneity. Extending the benchmark to support heterogeneous provider-controlled systems with different APIs, policies, and ranking objectives is an important direction for future work.
> > >
> > >
> > > **(3) Reframing contributions around what the controlled setup enables.** Just as BEIR (Thakur et al., 2021) fixes the retrieval model to isolate cross-domain generalization, and WebArena (Zhou et al., 2024) standardizes web environments to isolate agent planning, AgentWebBench fixes the website infrastructure to isolate coordination effects. We will **clearly state that all findings hold within this controlled context**, and that generalization to heterogeneous real-world settings remains an open question for future work.
> > >
> > >
> > > We hope the revised abstract above and the planned revisions demonstrate our commitment to narrowing the claims to match the setup. We will apply the same scoping throughout the introduction, contributions, and related work sections.

---

### Official Review · Reviewer_YeRc · 2026-03-13

**Soundness:** 2
**Presentation:** 2
**Significance:** 2
**Originality:** 2
**Overall Recommendation:** 3
**Confidence:** 3

**Summary:**

The authors propose AgentWebBench, a benchmark for evaluating how LLM agents gather information from multiple websites via tasks including web-scale search and document retrieval, document recommendation, question answering, and deep research report generation. Three interaction paradigms are proposed: embedding-based website tool selection (Tool_E), prompt-and-reasoning-based tool selection (Tool_P), and a multi-agent approach in which a "user" LLM issues queries to individual content agents, each representing a single website (Multi-Agent). Evaluations are conducted across seven LLMs on an offline dataset spanning the top 100 websites from ClueWeb22-B. Results show that Tool_P generally outperforms the other two approaches across tasks. The authors also analyze agent behavior and failure modes, reporting findings such as test-time scaling improving task performance, inter-model variation in tool-calling format compliance, and agent faithfulness to source content affecting downstream performance.

**Compliance With Llm Reviewing Policy:**

Affirmed.

**Final Justification:**

I thank the authors for their detailed rebuttals. While I do agree that W3: No retrieval model ablation, W4: Missing unified search engine baseline, and W5: Limited novelty due to existing data sources has been addressed, my concerns on W1: Disproportionate emphasis on Multi-Agent and W2: Multi-Agent implementation limitations remain. I have adjusted my scores accordingly.

**Key Questions For Authors:**

See the weaknesses above.

**Limitations:**

Yes.

**Strengths And Weaknesses:**

# Strengths
- The experimental setup is solid at a high level: the benchmark covers over 100 mainstream websites and 18M documents, simulating web-scale search and interaction. The chosen tasks (search, recommendation, QA, and deep research) are representative of real-world LLM web agent use cases, and the three interaction methods are plausible examples of how agents might engage with web content in practice.
- The detailed analysis of model performance, agent behavior, and failure modes is informative and practically useful for researchers and practitioners working on web LLM agents.
- The main results (Table 2) are intuitive and convincing. Tool_P is generally the strongest method for web agent interaction, as it provides the agent with direct content to multiple websites, and thereby offering the broadest context. Therefore, it is unsurprising that this configuration generally outperforms other alternatives.

# Weaknesses
- The paper places disproportionate emphasis on the multi-agent approach, which I find difficult to justify for two reasons. First, multi-agent interaction is not how LLM-based web agents are typically deployed in production today: popular approaches such as MCP combined with tool selection are much closer to Tool_P. Second, the multi-agent approach does not generally outperform Tool_P in the reported results, which further undermines the rationale for focusing on it in the paper.

- The specific implementation of the multi-agent approach raises additional concerns. Based on the prompts and examples in the appendix, the user agent can only issue general queries to content agents, which search the assigned website and return summarized results. Unlike a human participant in a multi-party conversation, the user agent cannot ask targeted follow-up questions or request specific clarification on the returned summaries. This effectively makes the multi-agent setup a worse version of Tool_P: the user agent lacks access to the original documents, and neither the user agent nor the content agents can reason over the full context of content retrieved from multiple websites. Under these constraints, it is unsurprising that the multi-agent approach underperforms.

- The performance of all three methods presumably depends heavily on document retrieval accuracy, yet the authors do not justify their choice of embedding retrieval model or conduct any ablation studies. A production-grade search engine based on both keyword heuristics and embeddings, such as Elastic Search, would be a more reasonable setup, and yet this is not implemented in the paper.

- A natural additional baseline method would be a single user agent with access to a unified search engine indexing all 100 websites, which allows searching all documents globally and eliminating the need for website / tool selection. This baseline method is currently not implemented in the paper.

- Given that all task queries and web crawl data are drawn from prior work (MS MARCO Web Search, ORBIT, DeepResearchGym, ClueWeb22-B) with no new queries or data collected by the authors, the paper's novelty is limited. In its current form, the multi-agent contribution does not constitute a meaningful contribution over prior work.

- The writing of the paper can be further improved. Key experimental details and conclusions are currently buried in a lengthy appendix. The main body of the paper should be self-contained and present all critical methodology and findings.

---

> ### Author Rebuttal · Authors · 2026-03-30
>
> > ### W1: Disproportionate emphasis on Multi-Agent
>
> AgentWebBench is a benchmarking contribution, not a proposal advocating Multi-Agent as the best approach.
>
> **(1) Multi-Agent coordination is an emerging reality.** We agree that MCP-based tool selection (similar to Tool_P) reflects current practice. However, the industry is moving toward agent-to-agent coordination: Google A2A, Anthropic MCP, and Microsoft NLWeb enable content providers to expose resources through agent-oriented interfaces (`Section 2`), and Anthropic has deployed multi-agent coordination in production (Anthropic Engineering Blog'25). Benchmarking this paradigm now is timely.
>
> **(2) The paper does not advocate Multi-Agent as the best approach.** The three strategies form a progressive evaluation framework isolating different factors. That Tool_P often outperforms Multi-Agent is itself a key finding: it reveals new challenges in decentralized coordination (e.g., summarization noise, `Sec. 5.5`) that current LLMs have not solved. Understanding these bottlenecks is the goal of this benchmark.
>
> **(3) Multi-Agent shows promise and extensibility.** It matches or outperforms Tool_P on QA for stronger models (`Table 2`), and the gap narrows with model scale. Its modular design lets researchers improve user and content agents independently (see Reviewer `tMi6` **Q2**), and enables studying adversarial content agents and provider-side ranking, which tool-based settings cannot.
>
> We will rebalance the framing in revision.
>
> > ### W2: Multi-Agent implementation limitations
>
> We clarify three factual points where the reviewer's characterization does not match our implementation.
>
> **(1) The user agent **`ALREADY`** have access to original documents.** In `Sec. 3.1`, each content agent returns both natural language summaries and full document contents. The user agent is not limited to summaries alone. What it cannot do is directly access the underlying corpus or bypass the content agent interface. This reflects real-world access constraints where platforms such as RedNote and healthcare systems (HIPAA) restrict external agents from freely crawling their content.
>
> **(2) The setup **`ALREADY`** support multi-turn adaptive interaction.** The user agent interacts with content agents for up to 15 turns (`Sec. 4`), maintaining full conversation history. It can re-query the same agent with refined follow-ups or switch to a different one based on earlier results (`Appendix F Case 1`).
>
> **(3) Content agents reason within their own website, which is the intended design.** This aligns with decentralized retrieval architectures such as FeB4RAG (SIGIR'24) and RAGRoute (arXiv 2502.19280), where each source maintains autonomous retrieval and the coordinator handles cross-source selection.
>
> Besides, the performance gap between Multi-Agent and Tool_P reflects new challenges introduced by decentralized coordination, such as summarization noise and interaction overhead (as analyzed in `Sec. 5.5` and `Appendix H`). These are valuable benchmark findings: they explain why Multi-Agent does not always outperform Tool_P and point to concrete areas where future Agentic Web research should focus.
>
> > ### W3: No retrieval model ablation.
>
> We choose dense retrieval because the consistency with ClueWeb22 and it also represents the mainstream approach in RAG systems. Here we also conduct additional experiments with BM25 and hybrid backends on web search with Gemini-3:
>
> | Retrieval | Tool_E | Tool_P | Multi-Agent |
> |---|---|---|---|
> | Dense (Ours) | 39.56 | 47.52 | 44.34 |
> | BM25 | 29.73 | 35.75 | 33.07 |
> | Hybrid | 37.59 |  45.35 | 44.94 |
>
> Results in N@3  (details in response to Reviewer `amui`). Across all backends, relative strategy rankings remain consistent and core findings hold.
>
> > ### W4: Missing unified search engine baseline
>
> The "Classical" baseline in our paper **`ALREADY`** is the unified search engine baseline (`Sec. 5`, `Table 2`). We will further clarify the description in revision.
>
> > ### W5: Limited novelty due to existing data sources.
>
> **(1) Our benchmark novelty lies in the research question.** Many high-impact benchmarks repurpose existing datasets for new evaluation dimensions, e.g.:
> - GLUE (Wang et al., 2018) unified existing NLU tasks.
> - BEIR (Thakur et al., 2021) assembled 18 existing retrieval datasets.
> - MTEB (Muennighoff et al., 2023) collected existing datasets.
>
> In each case the contribution was the **new evaluation lens, not new data**. AgentWebBench similarly introduces a new lens: multi-agent coordination in decentralized web environments, absent from existing benchmarks.
>
> **(2) This lens addresses an emerging reality.** The web is shifting toward a multi-agent ecosystem (Google A2A, MCP servers, Microsoft NLWeb). AgentWebBench is the first benchmark to formalize this setting, and its findings cannot be derived from any source dataset individually.
>
> > ### W6: Writing quality
>
> We will move key experimental details from the appendix into the revision.

---

> > ### Author Rebuttal · Reviewer_YeRc · 2026-04-04
> >
> > Thank you for the detailed and thoughtful rebuttal. I appreciate the authors' efforts to address my concerns. However, several concerns remain.
> >
> > > W1: Disproportionate emphasis on Multi-Agent
> >
> > > (1) Multi-Agent coordination is an emerging reality.
> >
> > I am not fully convinced that the examples cited (Google A2A, Anthropic MCP, and Microsoft NLWeb) effectively support this claim. While A2A is indeed designed for multi-agent interactions, MCP's primary purpose is to enable agents to interact with tools and data across multiple providers, not agent-to-agent communication. If anything, MCP would seem to support the Tool_P paradigm. Similarly, NLWeb appears to be primarily designed for embedding chat assistants into website GUIs, mainly serving human users.
> >
> > > (2) The paper does not advocate Multi-Agent as the best approach.
> > I appreciate the authors' clarification and appreciate their commitment to rebalancing the framing in the revision.
> >
> > > (3) Multi-Agent shows promise and extensibility.
> >
> > I find the argument here somewhat weak. Among the four tasks evaluated, Multi-Agent outperforms other approaches only on QA, and only for a limited set of models. I am also hesitant to agree that multi-agents "match or outperforms Tool_P on QA for stronger models." In particular, GPT-5 is widely regarded as a strong model, yet Tool_P outperforms Multi-Agent even on the QA task for this model.
> >
> > > W2: Multi-Agent implementation limitations
> >
> > > The setup ALREADY supports multi-turn adaptive interaction.
> >
> > After carefully re-reading both the rebuttal and Appendix F Case 1, as well as the remaining prompts in the appendix, I remain unconvinced on this argument. I was already aware that the user agent can issue multiple queries to the same content agent; however, these queries appear to be general in nature rather than targeted follow-ups. As suggested in my original review, human collaborators typically ask highly specific, context-aware follow-up questions. In contrast, the multi-agent queries in the paper read more like general Google searches independent of contexts. I also did not find any prompt instructions that explicitly encourage agents to issue targeted, context-aware follow-ups. For these reasons, I continue to believe that the multi-agent interaction is not yet implemented in a sufficiently meaningful way, which may partly account for the observed underperformance.
> >
> > The remaining aspects of W2, along with W3–W5, appear to have been largely addressed. I thank the authors for the additional experiments and explanations provided.

---

> > > ### Author Response · Authors · 2026-04-06
> > >
> > > > ### W1(1): Multi-Agent coordination is an emerging reality.
> > >
> > > We organize evidence spanning four layers.
> > >
> > > **(1) Community.** A survey of Agentic Web (arxiv:2507.21206) supports the growing adoption of multi-agent coordination in open-source communities. Besides, **OpenClaw**, the largest open-source AI agent project, includes built-in agent-to-agent communication primitives (session tools for agent-to-agent communication: `sessions_list`, `sessions_send`, `sessions_history`) and multi-agent routing as a core Gateway feature. Its ecosystem has already produced hundreds of recognizable multi-agent systems, such as Multi-Agent Content Factory, Multi-Agent Specialized Team, Autonomous Project Management, OpenClaw as Desktop Cowork (AionUI), and the OpenClaw Agents 9-Agent Kit.
> > >
> > > **(2) Protocols.** A dedicated protocol layer for agent-to-agent communication has emerged.
> > > - **A2A** (50+ enterprise partners) defines how agents discover, authenticate, and communicate with each other.
> > > - **ACP** (Agent Communication Protocol) by IBM provides an agent-to-agent interface adopted in the BeeAI platform.
> > > - **ANP** (Agent Network Protocol) addresses cross-network agent discovery.
> > >
> > > **(3) Frameworks.** Major AI companies have released a multi-agent orchestration framework. These are not research prototypes but production-grade SDKs: OpenAI Agents SDK (20.6k stars; "Handoffs" for inter-agent control), Microsoft AutoGen (56.7k stars) + Agent Framework (8.7k stars; A2A+MCP), Google ADK (built on A2A), CrewAI (role-based orchestration with A2A/ACP), LangGraph (graph-structured multi-agent workflows), and Anthropic's Orchestrator-Workers pattern.
> > >
> > > **(4) Enterprise products.** Multi-agent coordination has already been productized in enterprise products: Amazon Bedrock Multi-Agent Collaboration, Salesforce Agentforce, Google Agentspace, and SAP Joule are shipped products with paying customers, demonstrating that multi-agent coordination has moved from framework to enterprise software.
> > >
> > > > ### W1(3): Multi-Agent shows promise and extensibility.
> > >
> > > On QA accuracy, Multi-Agent outperforms Tool_P for five of seven models (Qwen3-4B, 14B, 80B-A3B, DeepSeek-V3.2, and GPT-5), with one tie (Gemini-3) and one loss (Qwen3-30B-A3B). For GPT-5 specifically, the comparison is mixed across metrics: Multi-Agent achieves higher accuracy (33.96 vs. 32.08) but lower F1 (22.78 vs. 24.99), so neither strategy clearly dominates. Additionally, in our experiments with external agents (response to Reviewer `tMi6` **W2**), Multi-Agent outperforms Tool_P on QA F1 across all three user-agent reasoning strategies (Ours, GPT Researcher, and Search-o1), confirming this advantage is not an artifact of a specific agent design.
> > >
> > > One of the benchmark findings is that Multi-Agent's advantage is task-dependent: it benefits tasks requiring iterative evidence gathering (QA) and shows comparable performance on synthesis tasks (deep research), while introducing overhead on precision-critical retrieval tasks (web search). Understanding when and why different coordination strategies succeed or fail is also the value of the benchmark.
> > >
> > > We agree with the reviewer's concern and will revise the paper to present this more carefully.
> > >
> > > > ### W2: The setup ALREADY supports multi-turn adaptive interaction.
> > >
> > > We thank the reviewer for reading our paper carefully. Below we explain why the prompt designed in this way is fair.
> > >
> > > **(1) The prompt is neutral to avoid biasing the evaluation toward any particular strategy, following consensus of agent design.** The user agent must balance **depth** (following up on the same website) and **breadth** (exploring new websites). Explicitly encouraging follow-up in the default prompt would bias evaluation toward one strategy. We chose a neutral prompt that follows consensus in agent design  (ReAct, Reflexion, etc.) and agent benchmarks: WebArena lists available actions without prescribing navigation strategy; AgentBench exposes interfaces without specifying usage patterns.
> > >
> > > **(2) The observation reflects current LLM capability, not a benchmark constraint.** The benchmark provides the full mechanism for targeted follow-ups: the user agent receives conversation history, can re-query the same content agent with a refined query, switch to a different content agent, or produce the final answer. The fact that LLMs still issue general queries is an empirical finding about their coordination ability, not a limitation of the benchmark design.
> > >
> > > **(3) Experiment of follow-up prompting.** On QA, Multi-Agent (Gemini-3), an added instruction encourages the agent to issue context-aware, targeted follow-up questions when appropriate. The results are:
> > >
> > > | Mode | Acc. | F1 |
> > > |---|---:|---:|
> > > | Neutral (ours) | 67.92 | 61.10 |
> > > | Follow-up | 66.04 | 58.12 |
> > > | Δ | -1.88 | -2.98 |
> > >
> > > This suggests that explicitly encouraging follow-up can over-bias the agent toward depth, reducing its flexibility to explore additional websites when broader coverage is more useful for the task.

---

### Official Review · Reviewer_tMi6 · 2026-03-25

**Soundness:** 4
**Presentation:** 3
**Significance:** 3
**Originality:** 4
**Overall Recommendation:** 3
**Confidence:** 3

**Summary:**

- Presents AgentWebBench, a new benchmark simulating decentralized, multi-agent web interaction across four tasks (search, recommendation, QA, deep research), using 100 websites and ~18.4M documents.
- Compares three coordination strategies (embedding match website search, llm reasoning website search, Multi-Agent) across multiple LLMs, showing that decentralized multi-agent coordination generally underperforms centralized retrieval.
- Agentic access concentrates web traffic to a small set of sources, test-time scaling (reasoning) improves reliability and performance, and effective coordination requires sufficient interaction and planning,
- User agents struggle with planning and answer synthesis, while content agents suffer from retrieval quality issues

**Compliance With Llm Reviewing Policy:**

Affirmed.

**Final Justification:**

After reading the rebuttal and the response between YeRc and the authors, I feel like the multi-agent system wasn't properly evaluated in this benchmark, thus limiting the takeaways from the benchmark.

**Key Questions For Authors:**

- Can the authors talk about how faithful is the simulated web is to the actual web environment? Can the benchmark capture enough of the real web?
- Are there any existing methods (i.e., different agent architectures) that can be used to evaluate on this benchmark? For example, individual tasks like Deep Research can be evaluate with a specific deep research agent but the benchmark is lacking evaluation on existing works.

**Limitations:**

See weaknesses.

**Strengths And Weaknesses:**

Strengths:
- One of the first benchmarks targeting multi-agent coordination in the emerging Agentic Web paradigm, addressing a clear gap beyond single-agent web benchmarks.
- Covers diverse tasks (search, recommendation, QA, deep research), multiple coordination strategies, and several LLMs, enabling broad and systematic analysis.
- Goes beyond standard metrics to study ecosystem effects (traffic concentration), efficiency (interaction budgets), and failure modes, providing actionable research insights

Weaknesses:
- The simulated environment (fixed corpus, synthetic agent interfaces) may not fully capture real-world web dynamics, API heterogeneity, or adversarial behaviors.
- Agent design is fixed, may be sensitive to retrieval infrastructure, prompt design, and coordination strategies, limiting generalizability of conclusions.

---

> ### Author Rebuttal · Authors · 2026-03-30
>
> > ### **Q1&W1: How faithful is the simulated web to the actual web?**
>
> Our simulated web balances realism with controlled evaluation.
>
> **(i) Corpus backed by established web benchmarks.** Our corpus is built on ClueWeb22 (Overwijk et al., 2022), developed by Carnegie Mellon University and Microsoft, is a large-scale web crawl corpus. It has been adopted in recent TREC web-scale evaluation tracks, such as TREC Interactive Knowledge Assistance Track and TREC Lateral Reading Track.
>
> **(ii) Diverse heterogeneous web environments.** The 100 websites with 18.4M documents (`Appendix L`) differ substantially in topic distribution, document quality, and domain specificity, creating heterogeneous coordination challenges.
>
> **(iii) Gaps are acknowledged and deliberately scoped.** The main gaps are: (1) static vs. dynamic content, (2) uniform vs. provider-specific retrieval, and (3) cooperative vs. adversarial agents. These are deliberate trade-offs to isolate coordination effects, consistent with WebArena (Zhou et al., 2024) and BEIR (Thakur et al., 2021). Our ananlysis (**W2**) retrieval ablation (response to Reviewer `amui`) confirms robustness across backends, and the modular design (**Q2**) supports per-website heterogeneity in future extensions.
>
> > ### **Q2: Can existing methods be evaluated on this benchmark?**
>
> Yes. We adapt existing deep research agents as user agent:
> - **GPT Researcher** (Assaf, 2023, 26.1k stars) is a well-known open deep research agent designed for both web and local research on any given task. It replaces the single-query search with a plan-and-solve pipeline: each search action first decomposes the query into multiple sub-queries via LLM, retrieves documents for each sub-query independently, then aggregates all sub-query results into a unified information summary through another LLM call before returning it as the observation.
> - **Search-o1** (Li et al., EMNLP'25) is agentic search-enhanced large reasoning models. It augments the multi-turn reasoning loop with a Reason-in-Documents module: when the agent issues a search action, the retrieved documents are not returned raw but first refined by an LLM call that extracts only the information relevant to the agent's current reasoning chain, reducing noise before feeding it back as the observation.
>
> Both methods replace only the retrieval/reasoning strategy while sharing the same backbone LLM and document index for fair comparison. We evaluate their performance on deep research and QA tasks:
>
>
> | Arch., Gemini-3 | Task | KPR↑ | KPC↓ | Acc.↑ | F1↑ |
> |---|---|---|---|---|---|
> | Multi-Agent (GPT Researcher) | Deep Research |  60.88 | 1.41 | - | - |
> | Multi-Agent (Search-o1) | Deep Research | 61.53 | 1.53 | - | - |
> | Multi-Agent (Ours) | Deep Research | 62.45 | 1.37 | - | - |
> | Multi-Agent (GPT Researcher) | QA | - | - | 62.26 | 54.30 |
> | Multi-Agent (Search-o1) | QA | - | - | 67.92 | 59.89 |
> | Multi-Agent (Ours) | QA | - | - | 67.92 | 61.10 |
>
>
> Both external agents successfully adapt to AgentWebBench and produce meaningful results, confirming that the benchmark supports diverse existing agent architectures beyond our implementation.
>
> > ### **W2: Fixed agent design limits generalizability**
>
> We **vary each agent component separately and observed that the core findings remain consistent**. Our agent design is a combination of three main components: (1) LLM backbone, (2) retrieval backend, and (3) agent reasoning strategy.
>
> **(1) LLM backbone.** We evaluate seven LLMs from Qwen3-4B to Gemini-3 using the same prompt templates. As shown in `Table 2`, relative performance patterns are consistent across models. The same strategy rankings hold regardless of model scale or family, suggesting the conclusions reflect properties of the coordination strategies rather than artifacts of a particular LLM.
>
> **(2) Retrieval backend.** Our retrieval ablation (response to Reviewer `amui`) replaces dense retrieval with BM25 and a hybrid method, verifying that findings hold across retrieval backends.
>
> **(3) User-agent reasoning strategy.** As shown in **Q2**, we adapt two external agents (GPT Researcher and Search-o1) with different reasoning strategies as the user agent. GPT Researcher uses a plan-and-solve pipeline with multi-query decomposition, while Search-o1 employs reasoning-guided document filtering. We evaluate them on the QA task and show F1 score:
>
> | Arch., Gemini-3 | Classical | Tool_E | Tool_P | Multi-Agent |
> |---|---|---|---|---|
> | GPT Researcher | 49.66 | 52.62 | 53.69 | 54.30 |
> | Search-o1 | 54.76 | 56.64 | 59.28 | 59.89 |
> | Ours | 56.15 | 51.92 | 60.26 | 61.10 |
>
> Across all three reasoning strategies, the **key findings from `Sec. 5.1` hold** regardless of user-agent design: Tool_P outperforms Tool_E (Finding 2), and Multi-Agent shows competitive performance on QA (Finding 4). This confirms that our conclusions reflect properties of agentic web rather than artifacts of our specific agent implementation.

---

### Decision · Program_Chairs · 2026-04-30

**Decision:**

Accept (regular)

**Comment:**

The paper introduces AgentWebBench, a benchmark for studying multi-agent coordination in an emerging Agentic Web setting. It is a timely and technically solid contribution with broad experimental coverage. Three reviewers highlight its novelty and potential impact as one of the first benchmarks in this space, while also appreciating the scale and diversity of tasks. However, concerns were raised about the realism of the simulated environment, the limited heterogeneity across content agents, and the framing and implementation of the multi-agent setup.

While there are some concerns about its current practical relevance and implementation, the overall consensus leans positive, with most reviewers acknowledging the benchmark’s value as a first step toward studying decentralized agent coordination. Taking these points into account, I'm inclined to the positive side, as the paper is likely to stimulate further research.